# Systematic assessment of prognostic molecular features across cancers

## Graphical abstract

## Highlights

- Dysregulation of biologically coherent gene modules predicts cancer survival

- Systematic discovery of highly prognostic DNA/RNA *cis*-regulatory modules

- Gene modules are more predictive of survival than single-locus aberrations

- Cancer modules predict survival beyond standard clinical features in current use

## Authors

Balaji Santhanam, Panos Oikonomou, Saeed Tavazoie

## Correspondence

st2744@columbia.edu

## In brief

Santhanam et al. present a systematic analysis of the prognostic potential of diverse coherent gene modules across cancer cohorts. Dysregulation of such prognostic cancer modules provide significant additional predictive power relative to common histopathological indicators and prominent genetic aberrations in common clinical use.

Santhanam et al., 2023, Cell Genomics 3, 100262
March 8, 2023 © 2023 The Author(s).

CellPress

## Article

# Systematic assessment of prognostic molecular features across cancers

Balaji Santhanam,[1,2,3,4] Panos Oikonomou,[1,2,3,4] and Saeed Tavazoie[1,2,3,4,5,*]

[1]Department of Biological Sciences, Columbia University, New York, NY 10027, USA
[2]Department of Systems Biology, Columbia University, New York, NY 10032, USA
[3]Department of Biochemistry and Molecular Biophysics, Columbia University, New York, NY 10032, USA
[4]Irving Institute for Cancer Dynamics, Columbia University, New York, NY 10032, USA
[5]Lead contact
*Correspondence: st2744@columbia.edu

## SUMMARY

Precision oncology promises accurate prediction of disease trajectories by utilizing molecular features of tumors. We present a systematic analysis of the prognostic potential of diverse molecular features across large cancer cohorts. We find that the mRNA expression of biologically coherent sets of genes (modules) is substantially more predictive of patient survival than single-locus genomic and transcriptomic aberrations. Extending our analysis beyond existing curated gene modules, we find a large novel class of highly prognostic DNA/RNA *cis*-regulatory modules associated with dynamic gene expression within cancers. Remarkably, in more than 82% of cancers, modules substantially improve survival stratification compared with conventional clinical factors and prominent genomic aberrations. The prognostic potential of cancer modules generalizes to external cohorts better than conventionally used single-gene features. Finally, a machine-learning framework demonstrates the combined predictive power of multiple modules, yielding prognostic models that perform substantially better than existing histopathological and clinical factors in common use.

## INTRODUCTION

Choice of therapies should be guided by accurate assessment of patient risk. Treatment decisions are often driven by histopathology-based observations, which have limited predictive value and suffer from inter-observer variability.[1–3] Precision oncology approaches seek to improve long-term patient outcomes by defining molecular dependencies of cancer progression that augment existing diagnostic and prognostic evaluations at the clinic.[4–7] Here, we systematically determined the prognostic potential of a variety of molecular observations to identify the most predictive features and compared their prognostic utility with clinical and histopathological features in common use. It is known that molecular changes that underlie oncogenic transformation can be informative of key clinical phenotypes such as therapeutic responsiveness, tumor aggressiveness, and patient risk.[8–11] Previous efforts have utilized aberrations at individual genomic loci to stratify patient risk (e.g., mutations or copy-number changes).[4,12] Generally, these single-locus genomic approaches may not capture higher-order dependencies that reflect activities of co-regulated processes, pathways, and regulatory networks. On the other hand, methods that utilize the activity of functionally relevant gene groups can capture coordinated dysregulation across genes and their association with clinical phenotypes.[13–24] These efforts have largely explored associations between perturbations in a limited set of prominent and curated cancer-relevant pathways with patient survival. However, the comparative and combined predictive potential of gene groups, single-locus aberrations, and conventionally used clinical features has not been systematically determined across cancers. Thus, a systematic effort to evaluate the prognostic capacities of genetic lesions and dysregulation of individual genes and functionally coherent groups of genes would not only expand the set of clinically usable tumor biomarkers but may help prioritize molecular assessments that provide optimal clinical utility in each cancer.

Here, we have developed a robust computational framework to determine the prognostic strengths of a variety of molecular features relative to conventionally utilized clinical factors such as tumor stage, age, and histopathology across a vast compendia of cancer cohorts from TCGA. We have used this framework to systematically quantify the prognostic potential of individual genes as conveyed by their mutation statuses, copy-number aberrations, and expression changes and found that transcriptomic assessments provided the strongest survival stratification in a majority of cohorts examined. Next, we curated a large set of biologically coherent gene groups (modules) based on a variety of features, including gene functions, biological processes, and co-expression based on DNA/RNA motif sequence features. Remarkably, we found that mRNA expression perturbations of gene modules provided significantly better survival stratification across the majority of TCGA cohorts compared with genomic perturbations of individual loci as well as other conventionally utilized clinical assessments. Furthermore, these

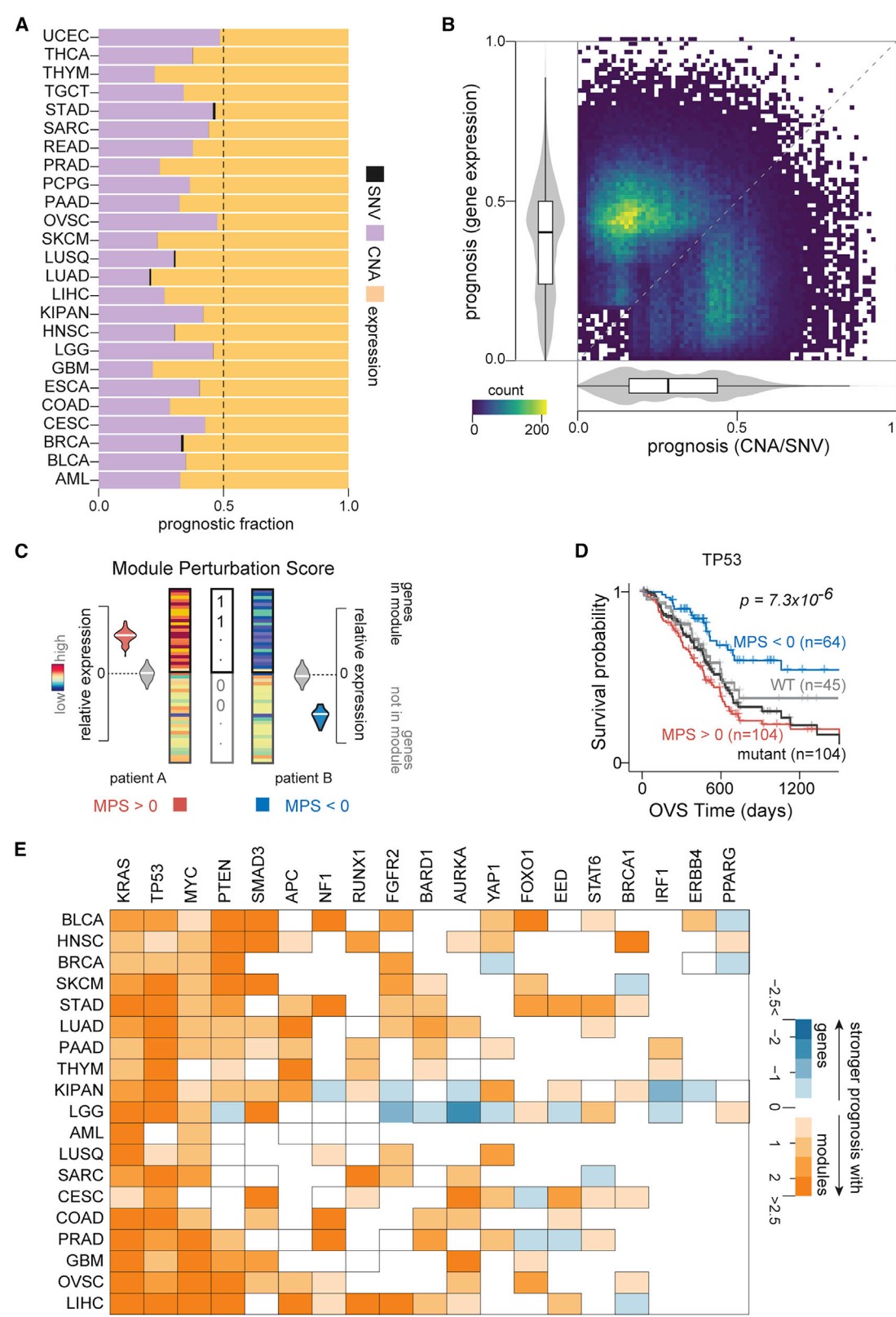

(legend on next page)

modules provided novel prognostic information compared with standard histopathologic assessments, prominent genomic aberrations, and compositions of immune cell types in the tumor microenvironment. Our analyses and conclusions provide the community with a powerful resource to generate clinically informative and interpretable models of patient risk, a critical foundation for precision oncology and therapeutic development.[1,7,10]

## RESULTS

### Gene expression is more prognostic than genomic aberrations at individual loci

We sought to determine the prognostic potential of individual gene measurements, including mutational statuses, copy-number aberrations, and gene expression changes across TCGA cohorts (STAR Methods). Our analyses on 8,620 patients from 25 cohorts in TCGA revealed many genes that were significantly associated with overall or progression-free interval survival ($p < 0.005$; Data S1) with a median number of 59 genes that were prognostic for both survival endpoints across cohorts. Overall, we found that the mutational status of a gene was a significantly weaker predictor of survival than copy-number aberrations (CNAs), as reported recently.[4] In fact, the expression of individual loci provided more prognostic utility than both mutations and CNAs (Figures 1A and S1). Indeed, in a majority of the cohorts tested, we also found that gene expression provided the strongest patient stratification for both overall survival (20 of 25 cohorts) and progression-free interval survival (21 of 24 cohorts) compared with CNAs or mutational aberrations (Figures 1B, S2, and S3), consistent with previous reports.[4,25]

Next, we focused on tumor-suppressors (n = 230) and oncogenes (n = 201) curated by OncoKB[27] based on their well-characterized roles as cancer drivers. Even among these cancer drivers,[27] we only found 8 genes (1.8%) based on mutational status, 148 genes (34.3%) based on copy-number changes, and 212 genes (49.2%) based on their expression levels to be prognostic for overall and progression-free interval survival in at least one cohort. Furthermore, 11 genes (5.2%) based on their expression and no genomic aberrations were prognostic in more than two cohorts. Only PTEN in low-grade glioma and BAP1 in

pan-kidney cohorts were prognostic across all three molecular features for either survival endpoint (Data S1). These results suggest that the clinical associations of these cancer drivers[27] are perhaps not adequately captured by their mutation statuses, CNAs, or expression changes alone.

### Gene modules associated with cancer drivers are more prognostic than mutational status

We hypothesized that gene modules coherently perturbed in the context of cancer driver aberrations may provide superior prognostic utility than the underlying single-gene perturbations. In order to test this, we first assembled 199 gene modules associated with cancer drivers[27] (22 tumor suppressors and 45 oncogenes) from the Molecular Signatures Database (MSigDB).[26] These modules were discovered on cells harboring perturbations in cancer genes and represent their transcriptional signatures (MSigDB).[26] To quantify the amplitude and direction of dysregulation in a module, we next defined a statistical measure based on signed mutual information[28] to quantify perturbations in each of these modules across patient primary tumor transcriptomes from TCGA (https://www.cancer.gov/tcga). We call this measure module perturbation score, which can be interpreted as the coordinated shift in the mRNA expression of a set of genes in a patient's tumor transcriptome (Figure 1C; STAR Methods). As expected, we found that module perturbation scores (MPS) were strongly correlated with mutations, CNAs, or expression changes for the majority of cancer drivers[27] (40 of 67) across the cohorts tested (Figure S4).

Next, we systematically explored whether perturbations in these modules stratified patients into groups with significantly divergent survival trajectories. These analyses identified modules associated with several cancer drivers[27] including TP53, KRAS, and PTEN, among others, to be informative of patient survival. Despite significant concordance between mutation status and perturbations in the TP53-associated module (hypergeometric $p = 8.8 \times 10^{-4}$), we observed that patients with pancreatic cancer stratified based on MPS had significantly more divergent survival trajectories (Kaplan-Meier [KM] $p = 2.5 \times 10^{-5}$; hazard ratio = 0.33) than patients stratified based on the mutation status of TP53 (KM p value = 0.28; hazard

---

**Figure 1. Expression changes are more prognostic than copy-number aberrations or gene mutations**

(A) Proportion of genes prognostic based on their copy-number aberrations (purple), mutation statuses (black), and expression changes (yellow) in each cohort (y axis) for both overall survival and progression-free interval survival.

(B) Comparisons between strengths of prognosis conveyed by gene expression (y axis) and genomic aberrations (x axis) across all cohorts is visualized as a heatmap scatterplot (density indicated). Within cohorts, absolute values of standardized significance (Wald statistic) of each stratification are scaled, and their distributions for prognosis utilizing gene expression (y axis) and genomic aberrations (mutation or copy number; x axis) are plotted. Only genes that are prognostic for at least one of the three features (mutation, copy number, or expression) are included.

(C) Schematic for the quantification of MPS. Mutual information is used to quantify the degree to which module membership is informative of gene expression levels in patient samples, which is then signed by the Pearson's correlation coefficient between them to yield MPS (STAR Methods).

(D) Kaplan-Meier (KM) plot shows patients with pancreatic cancer with positive perturbation scores (red) of a module corresponding to genes up-regulated in cell lines that harbor mutations at the TP53 locus relative to cell lines that are wild type for TP53 (MSigDB,[26] M2698; 198 genes) and have worse overall survival (OVS) than patients with negative perturbation scores (blue). Statistics (p value and number of samples in the two groups) are indicated (STAR Methods). KM plot of patients stratified based on TP53 mutation status in pancreatic cancer cohort is also shown (black and gray lines).

(E) The $\log_2$ ratio of the absolute standardized significance of modules associated with cancer drivers and measurements on genes encoding these cancer drivers (in rows) are visualized in 19 cancers from TCGA.

Standardized significance (Wald statistic) for individual genes were chosen to be the maximum from expression-, copy-number-, and mutation-based patient stratifications in each cohort. For the corresponding modules, the standardized significance scores were summarized using Stouffer's method (STAR Methods).

See also Figures S1–S8.

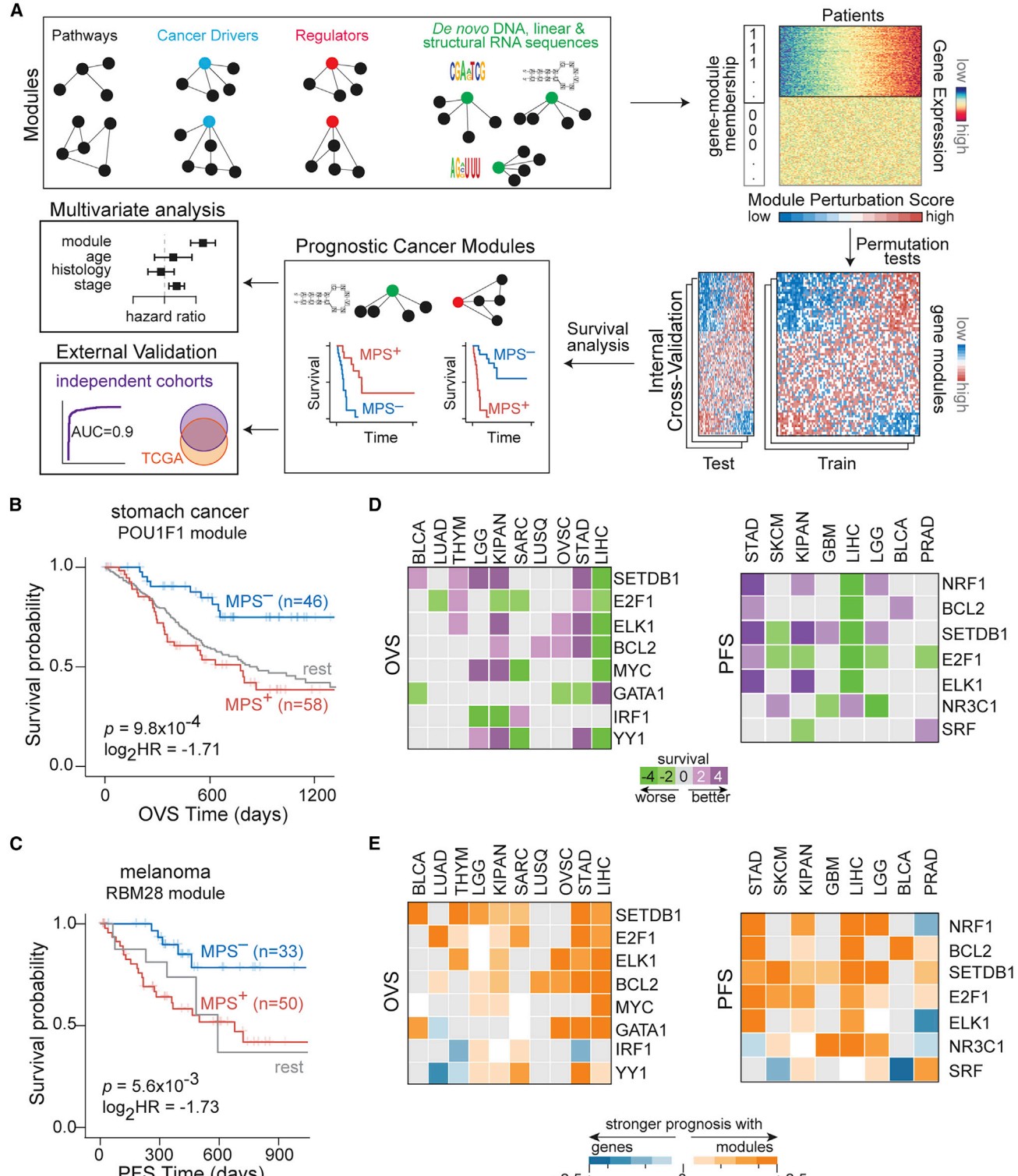

**Figure 2. Systematic discovery of prognostic cancer modules**

(A) Schematic for discovering prognostic cancer modules (PCMs). For every module, z scored transcriptome-wide data (heatmap) are systematically transformed into MPS across samples (heatmap; bottom). Patients with significant module activation (repression) have positive (negative) MPS values and correspond to transcriptomes in which genes in the module are activated (repressed) and labeled MPS$^+$ (MPS$^-$). Patient samples are stratified into MPS$^+$ and MPS$^-$ groups to quantify survival differences. Modules whose perturbations resulted in patient stratification with significantly different survival trajectories are considered to be

*(legend continued on next page)*

ratio = 0.76; Figure 1D). Remarkably, for ~90% of the cancer drivers,[27] patient survival stratifications based on module perturbations were superior to stratifications based on measurements at the corresponding individual loci across the cancers tested (Figures 1E and S5–S8). Taken together, these results suggest that the dysregulation of modules may be indicative of functional downstream consequences beyond direct observable genomic and transcriptional perturbations to upstream cancer drivers.[27]

### Systematic discovery of prognostic cancer modules

The superior prognostic utility of gene modules associated with cancer drivers motivated us to systematically discover other predictive gene modules across cancers. We thus expanded our scope broadly to include modules beyond oncogenic signatures. Our module catalog comprised ~5,000 pathways, gene ontologies, and ~2,000 putative targets of regulators including genes that harbor their binding sites and/or constitute their transcriptional targets (Figure S9A; Data S2; STAR Methods).[26,29–31]

We systematically quantified the dysregulation of each of these modules in every sample from TCGA (https://www.cancer.gov/tcga) as MPS (STAR Methods). We found that 25% of previously defined modules (Figure S9A) displayed distinct cohort-specific (Figures S9B and S9C) and tissue-specific (Figures S9D and S9E) perturbation patterns across the cohorts examined (Data S3; STAR Methods). We also sought to test if MPS can capture disease progression. To this end, we utilized cancer stage as a proxy for disease progression, identifying 3,221 modules in 12 cancers from TCGA (STAR Methods; Data S3; Figure S10A). We found that the perturbation scores of modules relating to mitosis, extracellular matrix, and RNA metabolism as well as transcriptional targets of polycomb group ring finger proteins BMI1 and PCGF2 were recurrently associated with disease progression in 3 or more cancers (Data S3). To test if our findings can generalize to different indicators of disease progression in independent patient groups, we utilized a cohort consisting of normal, polyps, primary tumor, and metastatic samples obtained from 342 patients with colonic neoplasms (GEO: GSE41258).[15,32,33] A majority of the modules discovered to be associated with colon cancer stage in TCGA were significantly associated with disease progression on this independent cohort. We identified modules

with substantial similarities to apoptosis and oxidative phosphorylation pathways (hypergeometric p = $10^{-11}$ and $10^{-237}$, respectively), consistent with observations originally made by Drier and colleagues[15] (Figures S10B–S10E). We also discovered potentially novel modules relating to mitochondrial organization, protein localization, and targets of zinc finger transcription factor PATZ1 to be significantly associated with colon cancer progression (Figures S10 and S11; Data S3). Clearly, further experimental validation is necessary to establish the *in vivo* functional roles of these modules in disease progression. Nonetheless, our results suggest that MPS can effectively capture biologically relevant phenotypes including disease progression.

Next, we quantified the ability of module perturbations to predict patient survival in individual cancers (schematic in Figure 2A). Modules whose perturbation scores stratified patients into activated ($MPS^+$) and repressed ($MPS^-$) groups with significant differential survival trajectories were defined as prognostic cancer modules (PCMs) (STAR Methods). We used TCGA as our primary discovery cohort due to (1) the diversity of cancer types analyzed, (2) the wealth of linked clinical, genomic, and molecular data on patients, and (3) the concordance between the distributions of pathologic stages in TCGA and the general population statistics in the United States in multiple cancers (Figure S12A).

Overall, among previously defined gene modules (Figure S9A), our analyses identified 443 PCMs (Figures S12B and S12C). These included modules associated with apoptosis, cell cycle, and DNA repair pathways, consistent with their well-established contribution to cancer progression (Data S4). We also identified ~180 known regulator-associated PCMs. Some examples included modules associated with microRNA (miRNA)-149, transcription factors ZHX2 and FOXF2, and the SRSF2 RNA-binding protein (Figures S13A–S13D). These findings are consistent with the known cancer-relevant roles of these regulators.[35–38] However, to the best of our knowledge, the prognostic values of their associated modules have not been previously determined.

An example of a novel regulator-based PCM is a module associated with the developmental transcription factor POU1F1, whose significant activation conferred worse prognosis in stomach cancer (KM p = $9.8 × 10^{-4}$; hazard ratio = 0.3; Figures 2B and S13E). To the best of our knowledge, neither POU1F1 nor its associated module have been previously implicated in

---

prognostic (STAR Methods). Confounding effects of conventional clinical and histopathological factors are controlled by jointly modeling effects of these covariates along with MPS on patient survival. The predictive performance of these PCMs is evaluated on independent external cohorts.

(B) Patients with stomach cancer with significant module activation ($MPS^+$; red) for genes harboring at least one instance of the binding site for POU1F1 (MSigDB,[26] M15591; 233 genes) have worse OVS than patients with significant module repression ($MPS^-$; blue).

(C) Patients with melanoma with significant module activation ($MPS^+$; red) for genes carrying at least one instance of RBM28 binding sites in their 3' UTRs (CISBP-RNA[30]; 1,595 genes) have worse progression-free interval survival (PFS) compared with samples with significant module repression ($MPS^-$; blue). For KM plot comparisons, statistics (median survival times, $\log_2$ hazard ratio, and p value) are indicated, and survival of the rest of the samples is shown in gray.

(D) Standardized significance of MPS-based patient survival (Wald statistic) of regulator-based modules are shown for OVS (left panel) and PFS (right panel). Regulator-based modules recurrently prognostic in 3 or more cancers are grouped together, and each row corresponds to the exemplar[34] module within a cluster (STAR Methods). For each module, patients in a cohort are stratified into $MPS^+$ and $MPS^-$ groups to quantify survival differences between the two groups. Positive (or negative) values indicate better (or worse) survival of patients in the $MPS^+$ group.

(E) The $\log_2$ ratio of the absolute standardized significance of module perturbations associated with regulators and measurements on their corresponding single genomic loci (in rows) are visualized for OVS (left panel) and PFS (right panel).

For single-locus measurements, standardized significance (Wald statistic) was chosen to be the maximum from expression-, copy-number-, or mutation-based patient stratifications in each cohort, and for their associated modules, standardized significance scores were summarized using Stouffer's methods (STAR Methods).

See also Figures S13–S15.

stomach cancer. Another example of a novel regulator-associated PCM is a module that corresponds to sequence-specific binding of RBM28, a component of the spliceosomal machinery. Significant activation of this module specified worse prognosis in patients with melanoma (KM p = 5.6 × 10$^{-3}$; hazard ratio = 0.3; Figures 2C and S13F).

In addition to cohort-specific prognostic modules, we also identified regulators whose MPS stratified patients into MPS$^+$ and MPS$^-$ groups with differential survival trajectories across multiple cancer cohorts (example in Figure S14). In fact, 57 and 70 regulator-associated modules were recurrently prognostic for overall survival and progression-free interval survival, respectively, in at least 3 cohorts (Figures S15A–S15C), which were in turn grouped into clusters whose exemplars are shown in Figure 2D (STAR Methods). These results suggest that while a sub-set of perturbations to regulators are prognostic in specific cancers, others are recurrently prognostic across multiple cancers, alluding to shared regulatory states underlying patient survival. Finally, as with the cancer drivers,[27] perturbations in regulator-associated modules improved patient stratification compared with expression changes, CNAs, or mutation statuses of the genes encoding these regulators (Figure 2E). These findings suggest that subtle and/or indirect modulation of regulators (e.g., through post-translational effects), which would not be captured by standard sequencing or immunohistochemical methods, may be sufficient for large-scale reprogramming of their targets, thus mediating significant effects on patient clinical trajectories.

## Dysregulation of *de-novo*-discovered *cis*-regulatory modules predicts patient survival

Our findings clearly establish that dysregulation of previously defined coherent gene modules convey substantial prognostic utility across cancers. However, we reasoned that gene expression dynamics across a large cancer cohort may point to coherently modulated genes with substantially higher relevance for disease progression across the cohort. Our known regulator-based gene modules above likely capture some of these changes. However, a majority of known regulators do not have associated target gene modules, and in cases where such modules exist, they are typically defined in cellular contexts that differ from those of patient tumors. We and others have shown that context-specific binding of regulators to *cis*-regulatory sequences in promoters or mRNA untranslated regions (UTRs) of target genes cause coordinated changes to their gene expression and that the underlying DNA/RNA *cis*-regulatory elements can be discovered by *de novo* sequence motif discovery.[39–41] We thus sought to systematically discover cancer context-specific *cis*-regulatory sequences that underlie gene expression perturbations across the TCGA cohort transcriptomes. Modules associated with these putative DNA/RNA sequence motifs would thus be coordinately regulated gene sets with potential importance in cancers. Using a *de novo* linear sequence motif discovery algorithm called FIRE,[42] we generated a systematic catalog of ~1,300 DNA and ~5,500 RNA putative regulatory sequence motifs that are significantly informative of tumor gene expression patterns across the TCGA cohorts (Data S2; STAR Methods). Since structural RNA regulatory elements can also play a major role in post-transcriptional regulation,[40,41] we

systematically discovered potential structural RNA motifs that are significantly associated with transcriptome dynamics (STAR Methods). To this end, we applied our structural RNA motif finder, TEISER,[40] to breast and liver cancer cohorts in TCGA to identify ~700 putative structural RNA regulatory motifs (Data S2; STAR Methods).

As expected, a sub-set of the short, *de-novo*-discovered patient-derived motifs were similar to binding sites of cancer-relevant regulators including ETS1, ELK1, FOS, JUN, and MAX (DNA elements) and HNRNPLL, RBM6, ELAVL1, miRNA-106, miRNA-525, and miRNA-329 (linear RNA elements) (TOMTOM[43] q <0.01; STAR Methods). However, for the majority of the *de-novo*-discovered motifs (>90%), we identified no significant matches to known binding sites of regulators (Data S5). For each motif, its associated module comprised genes that carry at least one instance of the motif in their regulatory regions (STAR Methods). Among these *de-novo*-discovered *cis*-regulatory motif modules, there were high degrees of overlap pointing to high inter-motif sequence similarities and/or co-occurrence of motifs. To minimize the redundancy between these modules, we grouped them into 1,050 module clusters with associated exemplars[34] for each (Data S5; STAR Methods). Taken together, the *de novo* discovery of DNA and RNA regulatory motifs revealed a large set of novel gene regulatory modules underlying patient transcriptome dynamics.

We assessed the prognostic potential of *de novo* discovered *cis*-regulatory modules in individual cancers independent of the cohorts in which they were discovered. We, thus, identified 157 DNA, 739 linear RNA, and 12 structural RNA-based non-redundant *cis*-regulatory modules to be highly prognostic for overall survival or progression-free interval survival (Figure S9; Data S2). A majority of these modules (~85%) remained significant even after accounting for confounding clinical factors using multivariate survival comparisons (Data S2; STAR Methods). We describe a few representative cases below.

We identified a DNA motif-based PCM whose activation confers worse prognosis in breast cancer (KM p = 9.6 × 10$^{-4}$; hazard ratio = 0.37; Figure 3A, S16A, and S16B). The motif resembled the E-box-like binding site for TFE3 (JASPAR ID MA0831.2[31]; TOMTOM[43] p = 1.68 × 10$^{-4}$; q = 0.1), which functions downstream of transforming growth factor β (TGF-β) signaling.[44] While TFE3 has been implicated in papillary renal cell carcinomas,[45] to the best of our knowledge, its role in breast cancer has not been reported. We also identified a DNA-based PCM whose activation specified better prognosis in prostate cancer (KM p = 2.3 × 10$^{-4}$; hazard ratio = 3.97; Figures 3B, S17A, and S17B). This motif did not share significant similarities with known binding sites of transcription factors (STAR Methods).

The vast majority of RNA motif-based PCMs did not match binding sites of known regulatory factors. Perturbations in one such module associated with a linear RNA *cis*-regulatory sequence conferred significant survival stratification of patients with stomach cancer, with activation of the module conferring worse survival (KM p = 8.6 × 10$^{-4}$; hazard ratio = 0.49; Figures 3C and S18A). This module was also effective at stratifying patients with stomach cancer with advanced stage tumors as well as older patients (Figure S18B). We also identified a structural RNA motif-based PCM whose significant activation specified

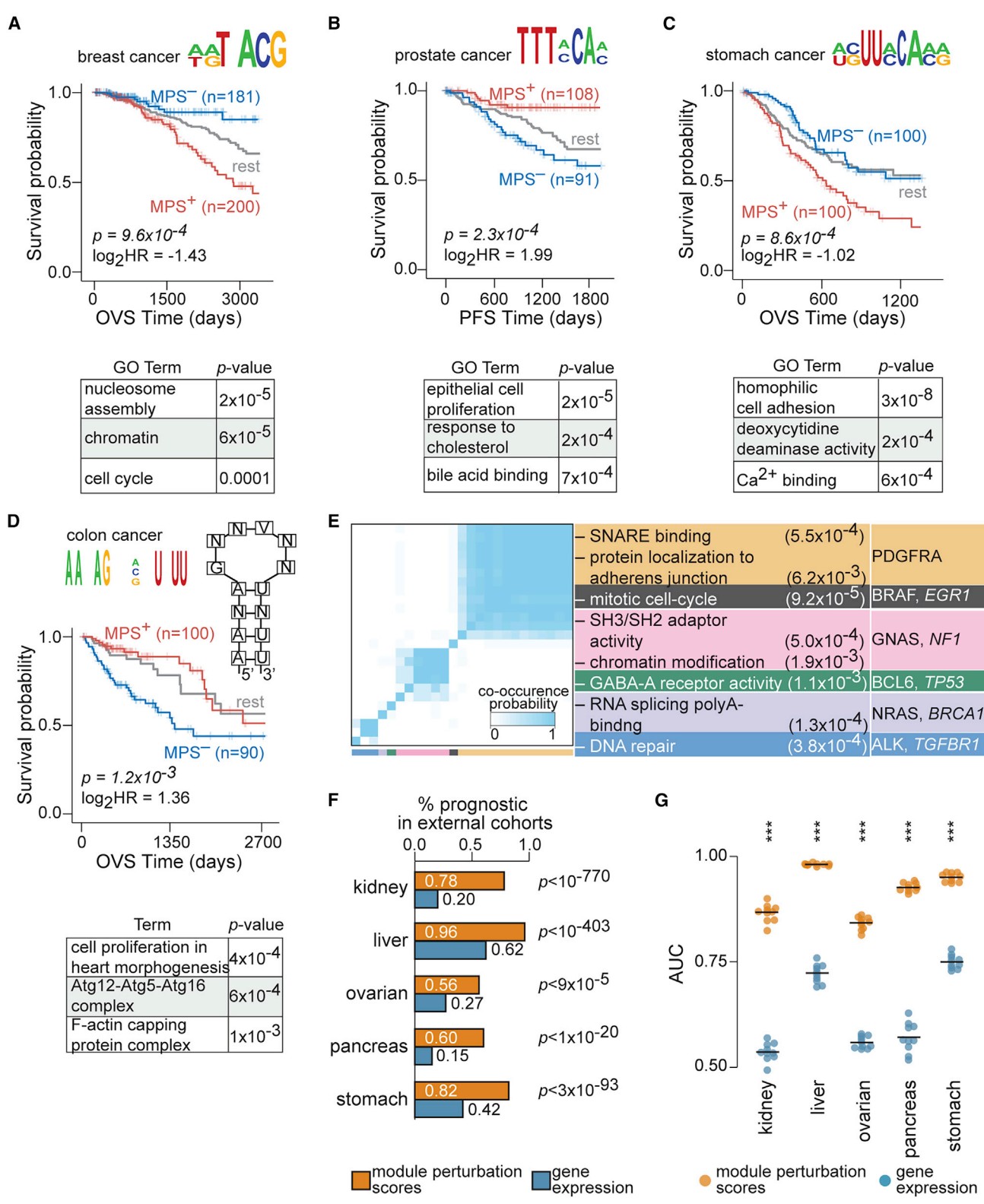

**Figure 3. Patient survival stratification based on *de-novo*-discovered PCMs and conserved prognosis in independent cancer cohorts**

(A) Patients with breast cancer with significant module activation (MPS+; red) for transcripts harboring at least one instance of the DNA motif HWRTNACGH (logo shown; 2,370 genes) within the first 1 kb of their promoters showed worse OVS than patients with significant module repression (MPS−; blue).

better prognosis in colon cancer (KM p = 1.2 × 10$^{-3}$; hazard ratio = 2.56; Figure 3D) independent of histological type, age, stage, race, microsatellite-instability status, and tumor location (Figure S19).

Out of all motif-based modules, 25 were prognostic in 3 or more cancer cohorts. Genes in these modules were enriched for processes known to be widely dysregulated in cancer including cell cycle, DNA repair, and chromatin organization, as well as known oncogenes and tumor suppressors[27] (Figure 3E). These examples are illustrative of many *de-novo*-discovered motif-based PCMs that are significantly informative of patient outcomes in individual cancer cohorts. For an expanded set of examples, see Figures S20 and S21 (full list in Data S4).

## Module perturbations are predictive of patient survival in independent cohorts

Our statistical criteria, including internal cross-validation on the TCGA cohorts, reduces the likelihood that the discovered PCMs may be overfit to one dataset. In order to provide another independent point of validation, we tested the ability of PCMs discovered across TCGA to stratify patient survival in independent external cohorts. We thus systematically quantified perturbation scores of modules and assessed their prognostic value on an independent set of over 800 patients[46,47] (STAR Methods). Differences between the TCGA and external cohorts in their demographic and histopathological compositions along with the limited number of external patient samples can impact consistent prognosis between the two datasets. Despite differences observed between the cohorts (Figures S22A and S22B), patterns of prognostic significance of modules in external cohorts were largely consistent with their results on TCGA (Figures S22C and S22D). In fact, PCMs discovered on TCGA were significantly over-represented in the set of modules discovered to be prognostic on the external cohorts, with 56%–96% of PCMs exhibiting consistent survival stratification in tissue-matched comparisons (Figure 3F). On the other hand, only a smaller fraction of genes based on their expression were consistently prognostic in the external cohorts (Figure 3F). Furthermore, the predictive performance of module perturbations to stratify patient survival in the independent cohorts was significantly better than gene expression (Figure 3G).

Taken together, these findings suggest that modules are superior at stratifying patients with significant survival differences in independent external cohorts compared with utilization of individual genes.

## Discovering PCMs in clinically relevant patient subgroups

First-line as well as subsequent therapeutic decisions in the clinic are made with considerations to patient's age, sex, stage, and histopathological characteristics of the tumors, as well as a few commonly assessed molecular features. Thus, the prognostic potential of module perturbations conditioned by prominent clinical and molecular factors may have important clinical utility. To this end, we sought to identify modules whose perturbations were prognostic in specific patient sub-groups by explicitly partitioning individual cohorts based on these features. Using a 3-fold cross-validation, we identified an additional set of ~1,400 non-redundant modules to be prognostic only within these *a priori*-specified sub-cohorts (Data S6). We describe a handful of these modules from a few categories below.

### Sex
A module associated with chromatin organization (Reactome R-HSA-489726; MSigDB[26]) was prognostic for overall survival only in female patients with glioblastoma multiforme and not in male patients (median survival difference of ~295 and ~70 days, respectively), likely identifying a group of high-risk female patients with glioblastoma multiforme (GBM) (Figure 4A). These results are suggestive of a novel, sex-specific role for chromatin biology in GBM disease progression.

### Histopathology
A module associated with homophilic cell adhesion (GO: 0007156; MSigDB[26]) was prognostic specifically in patients with triple-negative breast cancer and not in the full cohort (median survival difference of ~56 and ~17 months, respectively; Figures S23A and S23B). Another example was in patients with stage II/IIA/IIB breast cancer, where significant activation of a linear RNA motif-associated module conveyed worse prognosis even after accounting for confounding clinical factors (Figures 4B, S23C, and S23D).

---

(B) Patients with prostate cancer with significant module activation (MPS$^+$; red) for transcripts harboring at least one instance of the DNA motif DTTTMCAM (logo shown; 3,779 genes) within the first 1 kb of their promoters showed better PFS compared with patients with significant module repression (MPS$^-$; blue).

(C) Patients with stomach cancer with significant module activation (MPS$^+$; red) for transcripts harboring at least one instance of the linear RNA motif WSUUCAMR (logo shown; 1,872 genes) within the first 1 kb of their 3′ UTRs showed worse OVS compared with patients with significant module repression (MPS$^-$; blue).

(D) Patients with colon cancer with significant module activation (MPS$^+$; red) for transcripts harboring at least one instance of the structural RNA motif (logo and putative secondary structure indicated; 399 genes) within the first 1 kb of their 3′ UTRs showed better OVS than patients with significant module repression (MPS$^-$; blue). Select list of significant Gene Ontology terms enriched in each PCM are shown (bottom panel). For all KM plot comparisons, statistics (median survival times, log$_2$ hazard ratio, and p value) are indicated, and survival of the rest of the samples is shown in gray. For visualization, the time axis of KM curves is trimmed when the percentage of samples in MPS$^+$ or MPS$^-$ groups falls below 5%.

(E) *De novo cis*-regulatory PCMs that are recurrently prognostic in 3 or more TCGA cohorts co-cluster based on the similarities in their module memberships (modified Jaccard score; STAR Methods). Heatmaps show module co-clustering probabilities with six broad clusters (color key indicated) revealed by consensus clustering. Selected set of significant Gene Ontology terms associated with genes that are common to at least 75% of the modules in each cluster as well as prominent tumor suppressors (italicized) and oncogenes in this list are tabulated. p values indicating over-representation of GO terms (hypergeometric test) are indicated.

(F) Percentage of modules based on their perturbation scores (orange bars) or individual genes based on their expression (blue bars) that are consistently prognostic in tissue-matched independent cohorts (STAR Methods). p values indicate the significance of overlap for the modules (STAR Methods).

(G) Distributions of area under the receiver operating characteristic curves (AUC) are shown for MPS (orange) and single genes (blue) to predict patient prognosis on tissue-matched independent cohorts (STAR Methods).

The p values for comparisons between them (one-sided Mann-Whitney test ***p < 10$^{-5}$) are indicated.

See also Figures S16–S19.

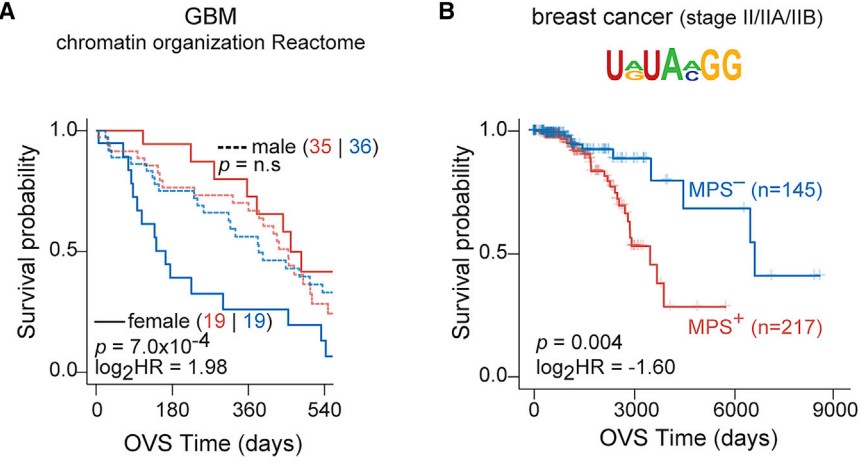

**A** GBM chromatin organization Reactome

**B** breast cancer (stage II/IIA/IIB)

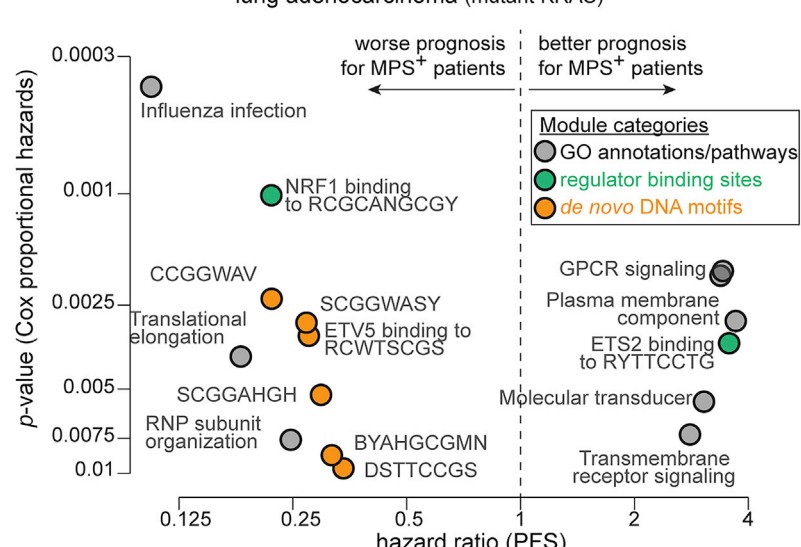

**C** lung adenocarcinoma (mutant KRAS)

**Figure 4. PCMs in clinically relevant patient sub-groups**

(A) Female patients with glioblastoma multiforme (GBM) with significant module activation (MPS+; red) for genes involved in chromatin organization (272 genes) showed better OVS than patients with significant module repression (MPS−; blue). Also shown are survival curves for MPS+ and MPS− male patients with GBM (dashed lines). Number of patients is indicated in parentheses.

(B) Patients with stage II/IIA/IIB breast cancer with significant module activation (MPS+; red) for transcripts harboring at least one instance of the RNA motif URUAMGGD (logo shown; 1,082 genes) within the first 1 kb of their 3′ UTRs showed worse OVS than samples with significant module repression (MPS−; blue).

(C) Volcano plot shows modules associated with Gene Ontology annotations and pathways (gray dots), regulator binding sites (green dots), and *de-novo*-discovered DNA-motif based modules (orange dots) that are clinically prognostic for PFS (hazard ratio: x axis; p value: y axis) in patients with KRAS-mutated lung adenocarcinoma.

(D) Patients with head and neck cancer and sarcoma with mutated TP53 and significant module activation (MPS+; red) for genes annotated to be involved in mRNA processing (243 genes) have worse OVS than patients with significant module repression (MPS−; blue). For the KM plots, statistics of the comparison (median survival times, $\log_2$ hazard ratio, and p value) are indicated.

See also Figures S23 and S24.

**D**

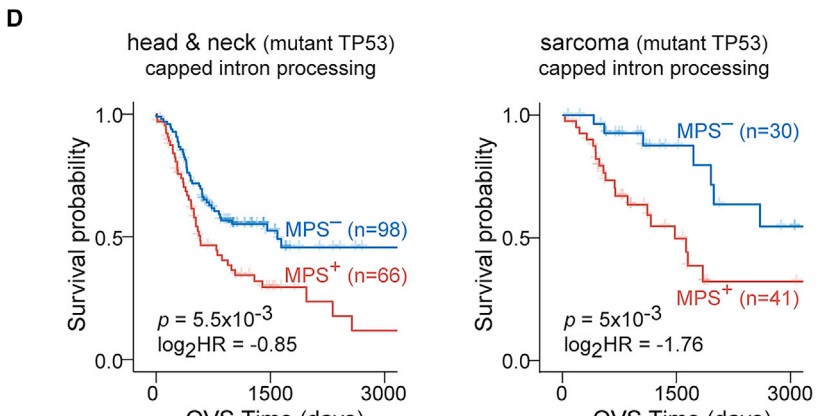

head & neck (mutant TP53) capped intron processing

sarcoma (mutant TP53) capped intron processing

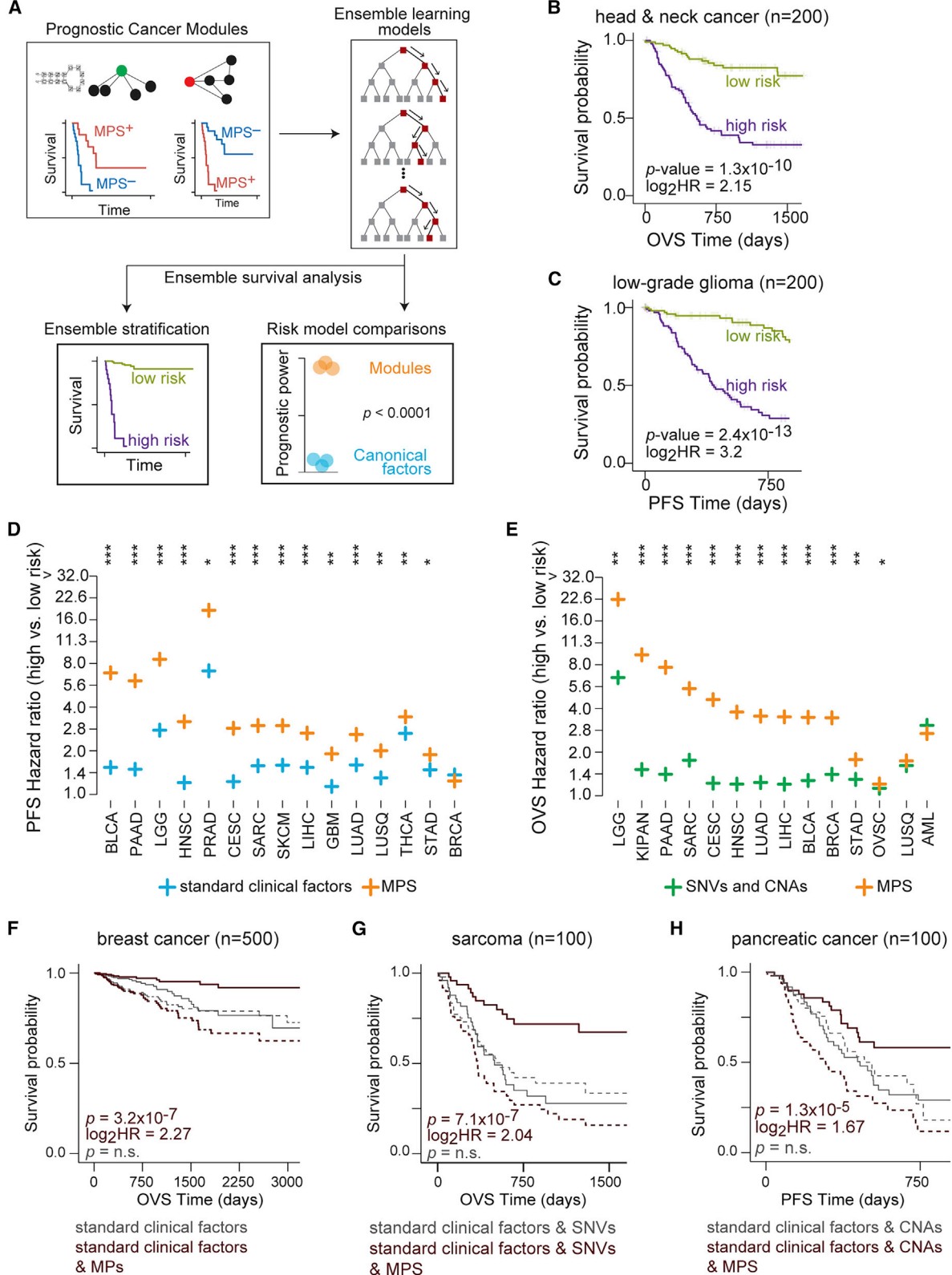

*(legend on next page)*

### Genomic aberration status

We identified a linear RNA motif-associated module to be prognostic in patients with breast cancer that harbor deep amplification of the MYC locus even after accounting for confounding clinical factors (Figures S24A–S24C). In another example, patients with colon cancer that harbor deletions of the ATP6V1B2 locus (component of vacuolar ATPase) with activation of genes involved in mitochondrion organization (GO: 0007005; MSigDB[26]) had favorable prognosis (Figure S24D). In KRAS-mutant lung adenocarcinomas, significant activation of NRF1 targets was associated with worse prognosis (Figure S24E), consistent with NRF1's role as a regulator of the proteasome pathway,[48] a key dependency in KRAS-mutant cancers.[49,50] We also established the prognostic value of modules associated with targets of ETV5 transcription factor, translational elongation, G protein-coupled receptor (GPCR) signaling, and ribonucleoprotein (RNP) sub-unit organization in KRAS-mutant lung cancers (Figure 4C). On the other hand, these modules had limited prognostic power in lung adenocarcinomas that were wild type for KRAS (Figure S24F). While further experimentation is essential to establish their functional roles, some of these modules, such as GPCR signaling, open up potentially novel avenues for therapeutic targeting in KRAS-driven lung cancers.[51] Finally, we also identified modules that were recurrently prognostic in sub-cohorts harboring aberrations at the same genomic loci in cancers of diverse tissues of origin (Data S6). For instance, mRNA splicing, which has previously been implicated in multiple cancers,[52,53] was prognostic in TP53-mutant sarcomas and head and neck cancers (example module from Reactome R-HSA-72203; MSigDB[26] shown in Figure 4D).

### PCM-based machine-learning models for integrated patient risk stratification

We sought to evaluate the combined prognostic power of all PCMs for a given cohort in a single predictive model. To this end, we utilized random survival forests[54] under 10-fold cross-validation to predict patient risk in individual cancers (schematic in Figure 5A). In a majority of the cohorts examined, we found that ensemble learning models trained on PCMs discovered in a cohort provided superior survival stratification of those patients compared with individual modules (Figure S25). For example, compared with the strongest individual PCM, models trained on multiple PCMs conferred an improvement in the median survival difference of ∼13.5 months in head and neck cancers (71 and 58 months, respectively; overall survival) and ∼43 months in low-grade gliomas (83 and 40 months, respectively; progression-free interval survival; Figures 5B and 5C). The predictive capacity of the random survival forest models was also validated on the external cohorts (Figure S26). To assess the context specificity associated with module perturbations, we used models trained on PCMs from one cohort to predict patient outcomes in every other cohort in TCGA (STAR Methods). Interestingly, we observed multiple instances of models trained on a cancer cohort conveying concordant prognosis in cancers of diverse tissues of origin (Figure S27), suggesting that patterns of module perturbations and their impact on patient survival may be conserved despite differences in their tissue contexts.

### PCMs add significant prognostic value to clinical factors in common use

Next, we sought to evaluate the power of modules, prominent individual genes, and clinically relevant histopathological factors to build comprehensive predictive models of patient survival in each cohort. We quantified contributions of PCMs to predict patient survival relative to typically used clinical and genomic features. First, we compared random survival forest models trained on PCMs discovered within a cohort with equivalent models trained on standard clinical factors alone. We found that models trained on PCM perturbations provided significantly superior patient stratification compared with equivalent models trained on standard clinical factors for ∼78% and ∼93% of cohorts based on overall and progression-free interval survival

**Figure 5. Models based on PCMs are predictive of survival beyond conventionally used clinical factors**

(A) Schematic for quantifying the combined predictive power of multiple PCMs and its relative strength compared with clinical factors in common use.

(B and C) KM plots show patient stratification based on risk predictions (high risk: purple; low risk: green) from a random survival forest model[54] trained on PCMs (see STAR Methods) in (B) head and neck cancer (OVS) and (C) low-grade glioma (PFS). Statistics of the survival comparisons and number of patients are indicated.

(D) Comparisons of random survival forest performance for predicting PFS in models trained on perturbation scores of PCMs (orange) and standard clinical factors (blue) across TCGA cancer cohorts. Each cross corresponds to the median hazard ratio from 10 different instances of 10-fold cross-validated models, and p values for comparisons between hazard ratios from the two models are indicated for each cohort (one-sided Mann-Whitney test ***p < $10^{-5}$; **p < $10^{-4}$; *p < 5 × $10^{-3}$).

(E) Comparisons of random survival forest performance for predicting OVS in models trained on perturbation scores of PCMs (orange) and prominent genomic aberrations (SNVs and CNAs) (green) across TCGA cancer cohorts. Each cross corresponds to the median hazard ratio from 10 different instances of 10-fold cross-validated models, and p values for comparisons between hazard ratios from the two models are indicated for each cohort (one-sided Mann-Whitney test ***p < $10^{-5}$; **p < $10^{-4}$; *p < 5 × $10^{-3}$).

(F–H) KM plots show patient survival stratification based on risk predictions (low risk: solid lines; high risk: dotted lines) from one of the 10 instances of random survival forest models. Predictions from this model trained using conventional clinical features and perturbation scores of PCMs are in dark red, while predictions of the comparable model without PCMs are in gray. Survival curves and associated statistics (p value and hazard ratio indicated) are for the instance with the largest difference in hazard ratios between the two random survival forest models. Survival comparisons are made using the same number of patients in each risk group, and total number of patients in each model (n) are indicated.

(F) In patients with breast cancer (n = 500), KM plots show patient stratification for OVS using standard clinical factors augmented by perturbation scores of PCMs.

(G) In patients with sarcoma (n = 100), KM plots show patient stratification for OVS using standard clinical features and SNVs augmented by perturbation scores of PCMs.

(H) In patients with pancreatic cancer (n = 100), KM plots show patient stratification for PFS using standard clinical features and CNAs augmented by perturbation scores of PCMs.

See also Figures S25, S28, and S30.

(Figures 5D, S28A, and S28B; STAR Methods). Likewise, we found that models trained on PCMs conferred superior patient survival stratification in ~81% of cohorts tested compared with prominent single-nucleotide variants and CNAs (Figures 5E and S28; STAR Methods).

We then sought to test if utilizing PCMs in conjunction with these conventionally used clinical factors provided additional predictive value. In the vast majority of cohorts tested, random survival forest models trained on both standard clinical factors and PCMs were significantly more prognostic than equivalent models trained on standard clinical factors alone (Figure S29). Similarly, models trained using SNVs or CNAs in combination with PCMs were significantly more prognostic than models trained only on SNVs (~78% of cohorts) or CNAs (~90% of cohorts) (Figures S30A, S30B, S31A, and S31B). Next, we tested if PCMs conveyed additional predictive power compared with standard clinical factors and prominent genetic aberrations combined. In fact, models trained on standard clinical factors and genomic aberrations combined with PCMs were significantly more prognostic than equivalent models trained without PCMs (Figures S30C, S30D, S31C and S31D). For instance, we found improved median survival difference of ~17 months (123 vs. 106 months) in breast cancers, ~25 months (32.5 vs. 7 months) in sarcomas, and ~22 months (23 vs. 0.6 months) in pancreatic cancer in models that included PCMs (Figures 5F–5H). Taken together, these findings suggest that perturbations in modules not only improve predictions of patient clinical outcomes but also provide additional prognostic information relative to standard histopathological factors and commonly utilized single-locus observations.

## DISCUSSION

We present a systematic computational framework to characterize how diverse molecular features and conventional clinical factors are able to predict patient survival across large cancer cohorts. We find that biologically coherent sets of genes (gene modules) provide a rich source of biomarkers with substantial prognostic utility, superior to single-locus observations (Figures 1 and 2). By utilizing DNA/RNA motif discovery in co-expressed genes, we have identified a large set of gene modules with significant prognostic value that promise to reveal novel biology with potentially significant contribution to cancer progression (Figure 3). While modules were chosen for their ability to provide biological context, computationally constructing modules, such that their perturbation scores maximize survival differences, could generate robust modules highly predictive of survival.

Our results suggest that MPS capture at least some of the molecular complexity underlying cancer, reinforcing the advantages of utilizing gene groups over single-locus observations.[55] The prognostic power of individual cancer modules motivated us to develop high-order machine-learning approaches that generate models by combining multiple PCMs with substantially improved survival prediction. In the vast majority of cases, these combined PCM models provide novel prognostic information that surpasses a variety of currently used histopathological and single-locus genetic aberrations (Figure 5). Interestingly, we found the contributions of MPS to survival predictions to be quite variable between cancers (e.g., low-grade glioma [LGG] and head and neck squa-

mous cell carcinoma [HNSC]; Figures S28–S30). One potential explanation is that only capturing genomic and/or transcriptomic states of tumors is unlikely to fully capture disease trajectories in all cancer contexts. Developing technologies and building compendia for multi-omic analyses of patient tumors with matched clinical data is likely to provide a more wholistic view of disease states and clinical outcomes.

The prominent roles inflammatory and immune cells play in oncogenesis and their importance in patient prognosis[56] inspired us to systematically compare survival predictions using frequencies of immune cell types[57,58] with predictions from PCM models. In more than 70% of cancers tested, models trained on PCMs provided significantly stronger patient stratification than models trained on immune cell-type frequencies (Figure S32).

Treatment decisions in the clinic are tailored to the histopathological and molecular features of tumors as well as patient-specific characteristics such as age and sex.[1,9,10] Within disease sub-groups, our approach reveals patients whose tumors may progress at a slower rate than the full cohort, as well as patients harboring malignancies that may progress more rapidly (Figure 4). Since information on treatments and the efficacy of these treatments is generally not available, the degree to which the prognostic potential of MPS captures inherently aggressive disease as opposed to reflecting therapy-resistant states that facilitate spread cannot be inferred from our results.

Our analyses reinforce the urgent need for expression profiling of tumor biopsies to become common.[1,6,8–10] Technical challenges with sample handling and storage as well as cost remain a major impediment for routine sequence-based analyses, suggesting that cost-effective transcriptomic sequencing at reduced depths may be advantageous.[6] In light of this, we expect the predictive power of module-based approaches, which are less reliant on noisy single-gene measurements, capturing patterns of coordinated gene expression changes instead, to be minimally impacted by information loss even when samples are sequenced at shallow depths.

### Limitations of the study

Firstly, while we believe our module-based approach would be highly beneficial for clinical decision-making, the specific predictions generated in this study may be affected by biases that inevitably exist in the underlying data distributions. Robust statistical standards and internal and external cross-validation certainly mitigate some of these concerns, providing confidence that our results may be generalizable. However, the clinical applicability of our predictions requires further validation on cohorts that closely resemble demographic (e.g., sex, age, etc.) and tumor characteristics (e.g., histopathology, tumor stage, grade, etc.) of the general population.

Secondly, molecular profiles of tumors utilized reflect steady-state genomic and transcriptomic states at the time of initial biopsy. Clearly, as patients receive treatments, tumors' microenvironment and molecular compositions may change in response to the interventions. As data on the effects of treatments and their outcomes become available, integrating them will help prioritize biomarkers for clinical utility.

While MPS can reflect gene expression perturbations within a module including non-uniform changes, coarse and non-specific

module definitions may affect the sensitivity of our approach. Thus, building fine-grained modules comprising genes with coherent changes in their expression could not only boost MPS but also improve downstream analyses. Finally, multivariate phenotypic attributes such as survival may exhibit non-monotonic dependencies with the patient's molecular characteristics. Supplementing our current framework with state-of-the-art statistical and machine-learning models may be essential to discovering these relationships.

## STAR★METHODS

Detailed methods are provided in the online version of this paper and include the following:

- KEY RESOURCES TABLE
- RESOURCE AVAILABILITY
  - Lead contact
  - Materials availability
  - Data and code availability
- METHOD DETAILS
  - Data sources and pre-processing
  - *De novo* discovery of *cis*-regulatory motifs
  - Relating regulators and *cis*-regulatory motifs
  - Previously defined gene modules
  - Minimizing redundancies between modules
  - Module perturbation scores
  - Prognosis of single-locus measurements
  - Prognosis of cancer-drivers and their modules
  - Cohort- and tissue-specific modules
  - Patient stratification and survival analyses
  - Identifying prognostic cancer modules
  - Modules associated with cancer stage
  - Validating prognosis of molecular features
  - Prognostic modules in patient sub-groups
  - Biological processes enriched in modules
  - Random survival forest models trained on PCMs
  - Prognosis of modules and clinical features
- QUANTIFICATION AND STATISTICAL ANALYSIS
- ADDITIONAL RESOURCES

### SUPPLEMENTAL INFORMATION

### ACKNOWLEDGMENTS

We thank all the members of the Tavazoie lab at Columbia University and Sohail Tavazoie, Dennis Hsu, and Benjamin Ostendorf at Rockefeller University for useful discussions and critical feedback. B.S., P.O., and S.T. are supported by the NIH/NCI award: 5R01CA257153.

### AUTHOR CONTRIBUTIONS

B.S, P.O., and S.T conceived the study, designed the experiments, analyses, interpreted the results, and wrote the manuscript. All authors read and approved the final manuscript.

### DECLARATION OF INTERESTS

The authors declare no competing interests.

### INCLUSION AND DIVERSITY

One or more of the authors of this paper self-identifies as a member of the LGBTQIA+ community.

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

## STAR★METHODS

### KEY RESOURCES TABLE

| REAGENT or RESOURCE | SOURCE | IDENTIFIER |
|---|---|---|
| **Deposited data** | | |
| TCGA SNV, CNA, RNA-seq and patient meta data | Broad Institute's Genome Data Analysis Center[59] | https://doi.org/10.7908/C11G0KM9 |
| TCGA patient meta data | Liu et al., 2018[60] | https://doi.org/10.1016/j.cell.2018.02.052 |
| ICGC transcriptome and patient data (kidney clear cell, liver, ovary, pancreas) | ICGC data portal[47] | Link (https://dcc.icgc.org/releases/release_28) |
| ACRG transcriptome and patient meta data (stomach) | Cristescu et al., 2015[46] | GSE62254 |
| Transcriptome and patient meta data (colon) | Martin et al., 2018,[32] Sheffer et al., 2009[33] | GSE41258 |
| **Software and algorithms** | | |
| R statistical computing environment (versions 3.2.3, 3.6.3) | R Core Team[61] | Link (https://www.r-project.org/) |
| FIRE | Elemento et al., 2007[42] | Link (https://tavazoielab.c2b2.columbia.edu/FIRE/) |
| iPAGE | Goodarzi et al., 2009[39] | Link (https://tavazoielab.c2b2.columbia.edu/iPAGE/) |
| TEISER | Goodarzi et al., 2012[40] | Link (https://tavazoielab.c2b2.columbia.edu/TEISER/) |
| TOMTOM | MEME suite[43,62] | Link (https://meme-suite.org/meme/doc/download.html) |
| **Other** | | |
| Resource website, code and data for generating main analyses of this paper | This paper | Resource website (https://tavazoielab.c2b2.columbia.edu/PCMs/landing_page.html) Code repository (https://github.com/balaji-srinivasan-santhanam/pcm) Data repository (https://zenodo.org/record/7418704) |

## RESOURCE AVAILABILITY

### Lead contact
Further information and requests for additional details and resources should be directed to the lead contact, Dr. Saeed Tavazoie (st2744@columbia.edu).

### Materials availability
This study did not generate new unique reagents.

### Data and code availability
This paper analyzes existing publicly available data. Accession numbers, digital object identifier(s) (DOI) and database versions are listed in the key resources table. Code used to generate key analyses and figures can be found at https://github.com/balaji-srinivasan-santhanam/pcm/. Any additional information required to reanalyze data reported in this paper is available from the lead contact upon request.

## Article

CellPress

## METHOD DETAILS

### Data sources and pre-processing

Analyses were carried out using the R statistical computing environment[63] (versions 3.2.3, 3.6.3). We used cancer cohorts analyzed through The Cancer Genome Atlas (TCGA) including Acute Myeloid Leukemia (AML), Bladder Urothelial Carcinoma (BLCA), Breast Invasive Carcinoma (BRCA), Cervical Squamous Cell Carcinoma and Endocervical adenocarcinoma (CESC), Colon Adenocarcinoma (COAD), Esophageal Carcinoma (ESCA), Glioblastoma Multiforme (GBM), Low-grade Glioma (LGG), Head and Neck Squamous Cell Carcinoma (HNSC), Pan-Kidney (KIPAN), Liver Hepatocellular Carcinoma (LIHC), Lung Adenocarcinoma (LUAD), Lung Squamous Cell Carcinoma (LUSQ), Skin Cutaneous Melanoma (SKCM), Ovarian Serous Cystadenocarcinoma (OVSC), Pancreatic Adenocarcinoma (PAAD), Pheochromocytoma and Paraganglioma (PCPG), Prostate Adenocarcinoma (PRAD), Rectum Adenocarcinoma (READ), Sarcoma (SARC), Stomach Adenocarcinoma (STAD), Testicular Germ Cell Tumors (TGCT), Thymoma (THYM), Thyroid Carcinoma (THCA) and Uterine Corpus Endometrial Carcinoma (UCEC). Molecular and clinical data were obtained from Broad Institute's Genome Data Analysis Center and from Liu et al.[59,60] We grouped tumors of stage I/IA/IB/II/IIA/IIB/IIC to be 'early' stage and tumors of stage III/IIIA/IIIB/IIIC/IV/IVA/IVB/IVC to be 'advanced' stage (AJCC pathologic stage). In cancers for which population-statistics (https://seer.cancer.gov/statfacts/, SEER 1975–2019) were available in tissue-matched cancers, we manually obtained proportion of early (*in situ* or localized) and advanced stage (regions) cancers in patients presenting with non-metastatic disease at time of diagnosis. We compared the proportions of early and advanced disease in TCGA and SEER cohorts for bladder (BLCA in TCGA), breast (BRCA in TCGA), colorectal (COAD, READ in TCGA), esophageal (ESCA in TCGA), kidney (KIPAN in TCGA), liver (LIHC in TCGA), lung (LUAD, LUSQ in TCGA), melanoma (SKCM in TCGA), oral (HNSC in TCGA), pancreas (PAAD in TCGA), stomach (STAD in TCGA) and thyroid (THCA in TCGA) cancers.

DNA-based molecular datasets included oncotator[64] annotated mutation data (SNVs) and germline-subtracted segmented somatic copy-number data (CNAs). RNA-based molecular data included microRNA data (reads per million) as well as sample-scaled quantile-normalized RNA sequencing data. Both gene-level and isoform-level data were used for transcriptome analyses. Additional RNA-seq datasets for liver, pancreatic, ovarian and clear cell kidney cancer cohorts were downloaded from the International Cancer Genome Consortium (ICGC) Data Portal (release 28). We only included cohorts with at least 50 patients for which both transcriptome and clinical data were available. Raw data were normalized by applying variance stabilizing transformation (R package DESeq2[65]). In ICGC's pancreatic cancer cohort (Canada), we only included tumor samples enriched by Laser Capture Microdissection with histological code 8500/3 (ductal carcinoma). For the ovarian cancer cohort, we restricted our comparisons to tumors with histological code 8441/3 (serous cystadenocarcinoma). Additionally, microarray data for stomach cancer patients were obtained from the Asian Cancer Research Group (ACRG; GSE62254). Robust Multichip Average (RMA) normalization was performed and probe identifiers were converted to gene symbols. Only unique identifiers were retained (n = 9419) for further analyses. Within individual cancers, standardized expression (*Z* score) per gene (or isoform) was calculated by subtracting the mean expression of the gene (or isoform) within the cohort divided by its SD in the cohort.

### De novo discovery of cis-regulatory motifs

First, we sought to infer linear DNA and RNA motifs underlying expression patterns in primary tumor and tumor-adjacent normal samples (when available). To this end, we used the linear motif discovery algorithm FIRE[42] for the *de novo* discovery of regulatory sequence elements in DNA and RNA. The space of sequences explored for identifying these motifs was restricted to one kilobase (1KB) immediately upstream of transcriptional start sites (TSSs) for DNA and 1KB immediately downstream of the coding region, encompassing the 3′ untranslated regions (UTRs) for RNA sequence elements. The presence (or absence) of these putative regulatory motifs was significantly informative of transcript abundance patterns. We analyzed z-scored isoform-level patient transcriptome data from each TCGA cohort using FIRE. In order to limit the influence and exploration of transcripts with low expression levels, we only included transcripts that on average were expressed above the 10th percentile of 100,000 randomly sampled data points across all samples within each cohort. NA values (obtained during *Z* score calculation) were made 0.

To unveil groups of coordinately expressed genes, we clustered primary tumor transcriptomes using CLARA (Clustering Large Applications)[66] implemented in the R package "cluster"[67] with two different cluster numbers. Number of clusters were specified as (i) $5 \times round\left(Ceiling\left(\frac{G}{3000}\right)/5\right)$ or (ii) $5 \times round\left(Ceiling\left(\frac{G}{500}\right)/5\right)$; $G$ is the number of transcripts; $Ceiling(x)$ is defined as the smallest integer that is not smaller than $x$ (R function 'ceiling'); $round(x)$ is defined to be the integer closest to $x$ (R function 'round'). We chose these cluster numbers to provide coarse and fine clustering (specified in (i) and (ii), respectively) of the data. Inputs to the FIRE algorithm included transcript identifiers (RefSeq IDs) along with their cluster identifiers (FIRE in discrete mode).

In cohorts that included normal samples from at least ten different patients (breast, colon, head & neck, kidney, liver, lung adenocarcinoma and lung squamous cell carcinoma, prostate, stomach and thyroid cancer cohorts), we additionally included comparisons between primary tumors and normal samples to run FIRE in continuous mode.[42] Inputs to FIRE included the log$_2$-ratio of transcript abundances in primary tumor compared to normal samples (difference between the means of their log$_2$-transformed abundances), grouped into 5, 10 or 25 equally populated bins.

In addition to identifying linear motifs, we applied TEISER[40] for the *de novo* discovery of putative structural RNA regulatory elements underlying patient transcriptomes. TEISER, like FIRE seeks to identify statistically significant relationships between the

presence or absence of pre-computed and optimized structural RNA motifs and transcript measurements.[39,40,42] To this end, we applied TEISER to breast and liver cancer cohorts in both discrete and continuous modes (described above).

In total, our analyses identified 7637 unique putative regulatory elements across the three categories with 1372 DNA motifs, 5555 RNA linear motifs and 710 RNA structural motifs. For every motif identified through the FIRE and TEISER analyses, we defined a module associated with the motif. We first defined transcripts that harbored instances of the motif in their promoters (DNA motifs) or 3′UTRs (RNA motifs). This collection of transcripts defined the motif's regulon and genes corresponding to these transcripts were defined to be the module.

### Relating regulators and *cis*-regulatory motifs

We quantified similarities between every *de novo* derived motif and the binding preferences of known regulatory factors. We applied TOMTOM to cross-reference these motifs against curated databases of binding preferences of regulatory factors (MEME version 5.0.5).[43,62] As input to TOMTOM, we used the IUPAC sequence of the motif (i.e., query) by converting it to a meme-formatted motif (iupac2meme function in MEME). We specified "JASPAR_CORE_2016_vertebrates.meme" (TFs) as the reference database to quantify similarities between these curated DNA regulatory elements and each query DNA motif. Similarly, for RBPs, we selected the database "Ray2013_rbp_Homo_sapiens.meme" and "Homo_sapiens_hsa.meme" for miRNA seeds as the reference to quantify similarities between these curated RNA regulatory elements and each query RNA motif. Using Pearson's correlation between the query and reference motifs (flag 'dist'), we required a minimum overlap of 5 (flag 'min-overlap') at an *E*-value threshold of 10 (flag 'thresh') and a *q*-value threshold of 0.01 to identify significant matches. Consistent with the paucity of validated structural binding elements, we found no significant matches to *de novo* identified RNA structural regulatory elements.

### Previously defined gene modules

In addition to *de novo* derived *cis*-regulatory motif modules, we also utilized a large collection of signatures curated through the Molecular Signatures DataBase (MSigDB version 7.0), assembling a more complete catalog of modules. These module definitions were obtained as unique NCBI Entrez gene identifiers that were then converted to gene symbols. These included 184 oncogenic and 796 motif signatures as well as 5052 Gene Ontology terms, pathway definitions. An additional set of 1149 regulator-based motif modules were sourced from CISBP, CISBP-RNA and JASPAR[29–31] databases. The vast majority of these previously defined gene modules are defined in cellular contexts with limited similarities to patient-derived primary tumors.

### Minimizing redundancies between modules

We sought to identify the extent of redundancy between individual modules within module categories including FIRE DNA motifs, FIRE RNA motifs (linear), TEISER RNA motifs (structural) as well as Gene Ontology and pathway terms. We defined a similarity measure to quantify the extent of overlap between two modules (i.e., sets of genes comprising the two modules) based on the Jaccard index, and defined as $S(A,B) = (G_A \cap G_B)/\min(G_A, G_B)$, where $G_K$ are genes in module $K$. Within each module category (listed above), we applied Affinity Propagation (R package 'apcluster')[34,68] on the matrix of similarity measures of all module pairs in that group to identify 209 module clusters (exemplars) for FIRE's DNA modules, 824 for FIRE's RNA modules, 17 for TEISER's RNA modules and 216 for Gene Ontology terms and pathways. The exemplar modules identified through Affinity Propagation were used to label all modules in a cluster.

### Module perturbation scores

We quantized standardized expression in individual samples into $N_e = 10$ equally populated bins. Each module is represented as a binary vector with 1 assigned to genes belonging to the module and 0 to rest of the genes, signifying the module membership vector. We calculated the mutual information between individual primary tumor transcriptomes and module membership vector to quantify the dependence between them:

$$I(module; expression) = \sum_{i=0}^{1} \sum_{j=1}^{N_e} P(i,j) log \frac{P(i,j)}{P(i)P(j)}$$

where $P(i,j) = C(i,j)/N$, $P(i) = \sum_{j=1}^{N_e} P(i,j)$, $P(j) = \sum_{i=0}^{1} P(i,j)$, $C(i,j)$ is the joint-counts table and $N$ is the total number of genes in each sample. The joint-counts table has 2 rows and $N_e$ columns. $C(1,j)$ indicates the number of genes in the module and in the $j^{th}$ expression bin, while $C(0,j)$ indicates the number of genes that are not part of the module and are in the $j^{th}$ expression bin.

These mutual information values were subjected to extensive randomization tests and only statistically significant scores were considered for further analyses, unless specified otherwise. To assess the significance of each mutual information score, we defined a null distribution by calculating mutual information between randomized primary tumor transcriptome and module membership vectors (1000 randomizations). The mean ($\mu$) and SD($\sigma$) of this empirical null distribution were used to calculate a *Z* score for the mutual information value. This *Z* score was considered significant if the mutual information value was within the top 1% of the values in null distribution. Negative *Z* score values were made 0. Finally, the sign of the Pearson correlation coefficient (*r*), between module membership vector and standardized expression of individual patient sample was used to sign the mutual information values to obtain the module perturbation score (MPS) for a given module and an individual patient's sample:

$$MPS(module; sample) = sgn(r) \cdot H\left(\frac{I(module; sample) - \mu}{\sigma}\right)$$

where $H$ is the Heaviside step function.

Thus, MPS indicates the module's perturbed activity in an individual patient transcriptome within a cancer cohort. Patients with significant module activation (repression) correspond to transcriptomes in which genes in the module are relatively up-regulated (down-regulated) and labeled MPS$^+$ (MPS$^-$). A module with a positive MPS score (MPS$^+$) in a patient (labeled significant module activation), corresponds to relative up-regulation of genes in the module in that patient transcriptome. Likewise, a module with a negative MPS scores (MPS$^-$) in a patient (labeled significant module repression), corresponds to relative down-regulation of genes in the module in that patient transcriptome. Different choices for number of bins $N_e$ (6, 10, 15) did not significantly impact downstream analyses.

### Prognosis of single-locus measurements

In each cohort for every gene, we compared survival trajectories (overall survival and progression-free interval survival) of patients stratified into groups with no detected mutations in that gene and those that harbor mutations in it. For copy-number aberrations (germline-subtracted segmented somatic copy-number data) and expression changes (z-scored gene expression data), patients were stratified into groups with relative amplification/deletion of gene copy-number and relative activation/repression of gene expression to quantify survival differences in each cohort. Only genes that had more than 15 patients in each group were included in these analyses. For comparisons between these three features, survival differences at a significance level of 0.05 (Cox proportional hazards model) were deemed prognostic.

### Prognosis of cancer-drivers and their modules

To compare patient stratification based on module perturbations and measurements on individual genes, we focused on the 199 modules from MSigDB[26] associated with 45 tumor-suppressors and 22 oncogenes from OncoKB.[27] For each module, their perturbation scores were calculated as described above. Thresholds based on the significance of z-scored mutual information values were not enforced. Patients were stratified into groups with positive and negative module perturbations to quantify survival differences between them in each cohort and for each significant module in every cohort, the absolute standardized significance scores (Wald statistic from Cox model; p < 0.05) were computed. Standardized significance scores for modules associated with each cancer-driver[27] in every cohort was aggregated using Stouffer's method. For measurements on the individual genes encoding the cancer-drivers,[27] patients were stratified as described above and we chose the strongest of the three stratifications, based on their absolute standardized significance scores (Wald statistic from Cox model).

### Cohort- and tissue-specific modules

We identified modules that exhibit substantial levels of dysregulation across patient samples within individual cohorts. We selected cohort-specific modules such that they have high mean absolute module perturbation scores (top 50%) as well as low variance (bottom 40%) within individual cohorts while treating MPS$^+$ and MPS$^-$ scores separately. We restricted our analyses to previously defined modules across 25 TCGA cancer cohorts, identifying 1978 modules with cohort-specific perturbation patterns. The patterns of perturbations remain largely unchanged when the complete catalog of modules was utilized.

Additionally, we identified modules that exhibited significantly differential perturbation patterns in primary tumor samples compared to patient-matched tumor-adjacent normal samples. We used z-scored transcriptome data from primary tumors and their matched normal samples and calculated module perturbation scores for all the modules in these samples. Modules whose perturbation scores were significantly perturbed in at least 75% of samples and have opposite signs in primary tumor compared to normal samples were labeled tissue-specific modules (i.e., MPS$^{tumor} \cdot$ MPS$^{normal}$ < 0). Our analyses discovered modules differentially perturbed in tumor-normal contexts in 10 cohorts: bladder cancer (313 modules, 19 patients); breast cancer (17 modules, 112 patients); colon cancer (610 modules; 26 patients); esophageal cancer (225 modules; 11 patients); head & neck cancer (20 modules, 43 patients); liver cancer (107 modules, 50 patients); lung adenocarcinoma (55 modules, 58 patients); lung squamous cell carcinoma (376 modules, 50 patients); rectal cancer (1450 modules, 6 patients); stomach cancer (22 module, 32 patients) and uterine endometrial cancer (493 modules, 7 patients).

### Patient stratification and survival analyses

For patient stratification, module perturbation scores were sorted and survival differences were quantified between patients with significantly activated (MPS$^+$) and significantly repressed (MPS$^-$) module perturbation scores using the 'survival' package in R.[69] The number of patients used in the MPS$^+$ and MPS$^-$ groups were determined based on the size of the cohort (n): AML (n = 149)- 25 patients per group, 50; BLCA (n = 401)- 25, 50, 75, 100, 150, 200; BRCA (n = 1061)- 25, 50, 75, 100, 150, 200, 250, 400, 500; CESC (n = 290)- 25, 50, 75, 100; COAD (n = 275)- 25, 50, 75, 100; ESCA (n = 183)- 25, 50, 75; GBM (n = 151)- 25, 50, 75; LGG (n = 510)- 25, 50, 75, 100, 150, 200, 250; HNSC (n = 519)- 25, 50, 75, 100, 150, 200, 250; KIPAN (n = 874)- 25, 50, 75, 100, 150, 200, 250, 400; LIHC (n = 360)- 25, 50, 75, 100, 150; LUAD (n = 480)- 25, 50, 75, 100, 150, 200; LUSQ (n = 488)- 25, 50, 75, 100, 150, 200; SKCM (n = 99)- 25; OVSC (n = 283)- 25, 50, 75, 100; PAAD (n = 177)- 25, 50, 75; PCPG (n = 178)- 25, 50, 75; PRAD (n = 486)- 25, 50, 75, 100, 150, 200; READ (n = 92)- 25; SARC (n = 257)- 25, 50, 75, 100; STAD (n = 383)- 25, 50, 75, 100, 150; TGCT (n = 134)- 25, 50; THYM (n = 119)- 25, 50; THCA (n = 500)- 25, 50, 75, 100, 150, 200; UCEC

 **CellPress**

**Cell Genomics**
Article

(n = 171)- 25, 50, 75. Additionally, for every module we also compared survival differences between all patients labeled MPS$^+$ and MPS$^-$. All survival comparisons were performed for both overall survival and progression-free interval survival times. For the AML cohort, only overall survival endpoint is defined.[60] We obtained the log rank test p values, Cox proportional hazards model p values, their associated z-scores from Wald test and hazard ratios (HR). For each comparison, we generated a null distribution and obtained false-discovery rates (FDRs) of the log rank test p value by randomly sampling similar sized patient groups (procedure repeated 100,000 times). Additionally, we also obtained false-discovery rates by randomly switching group labels from the stratification for individual patients. Survival curves were visualized using the 'survminer' R package version 0.4.8.[70] For visualization, survival curves were truncated when less 5% of at risk patients remained when specified.

### Identifying prognostic cancer modules

To minimize false discovery in patient stratification and survival, our analyses were performed with 3-fold cross validation. The transcriptome data collection in each cancer cohort was split 3-ways (2/3: training and 1/3: test) and thus, every sample appeared in a test set exactly once. Training and test datasets were each standardized as described above. For genes in the test set, their z-scores were calculated using their means and standard deviations in the training set. The module perturbation scores for all modules in our catalog were computed as described above for patient samples in the training and test sets. Samples were ranked by absolute non-zero MPS in both MPS activated (MPS >0) and repressed groups (MPS <0) to define MPS$^+$ and MPS$^-$ sample groups in the training set. We chose a fixed number samples from each group (numbers specified for each cohort above) and quantified the survival differences between them. From each training set, we propagated the boundary conditions for the activated (least positive value of MPS) and repressed (most negative value of MPS) sample groups to their respective test sample sets and categorized test set samples into MPS$^+$ and MPS$^-$ groups accordingly. We concatenated the three test sets to reconstitute the full cohort carrying the propagated MPS$^+$ and MPS$^-$ labels. On this full set, we quantified survival differences between the labeled MPS$^+$ and MPS$^-$ groups for the same fixed number of samples used on the training sets. We defined a module to be prognostic in a cohort when survival differences were significant (i) in each of the 3 training splits (log-ratio rank test FDR <0.05), (ii) in the label-propagated test set (log-ratio rank test FDR <0.01), and, (iii) in the full cohort (without training/test split) with the same fixed number of samples (log-ratio rank test FDR <0.01, Cox proportional hazard p value <0.05). These modules were defined to be prognostic cancer modules (PCMs).

A potential confound of the clinical significance of module perturbation scores could be due to correlations with clinical and histopathological factors, such as tumor stage, histological type, receptor status etc. To address this, we performed multivariate survival comparisons by jointly modeling the effects of these confounding factors in addition to module perturbation scores on patient survival using the Cox model and results were visualized as forest plots (R function ggforest; survminer package version 0.4.8[70]).

### Modules associated with cancer stage

Cancer stage information was available for 15 cancers from TCGA. Primary tumors of stage I/IA/IB were grouped to be 'stage I', tumors of stage II/IIA/IIB/IIC to be 'stage II', tumors of stage III/IIIA/IIIB/IIIC to be 'stage III' and tumors of stage IV/IVA/IVB/IVC were grouped to be 'stage IV'. We modified our computational pipeline for discovering PCMs in order to identify modules whose perturbation scores are associated with disease progression, using cancer stage as a proxy for disease progression. After calculating perturbation scores on the full catalog of modules as described earlier, we chose a fixed number samples from MPS$^+$ and MPS$^-$ groups (numbers specified for each cohort above). We then tested for preferential enrichment/depletion of either MPS group across the 4 discretized stages using Fisher exact test on the training and test datasets. We used the R function 'fisher.test' on the 4 × 2 contingency tables using default parameters with the flag 'simulate.p.value' set to TRUE to speed up calculations. We defined a module to be a progression-associated module in a cohort when enrichment/depletion patterns were significant (i) in each of the 3 training splits (Fisher exact test p value <0.05), (ii) in the label-propagated test set (Fisher exact test p value <0.01), and, (iii) in the full cohort (without training/test split) with the same fixed number of samples (Fisher exact test p value <0.01).

To validate modules associated cancer stage on an independent external cohort, we utilized a dataset consisting of 342 patients with colonic neoplasms from whom normal tissue, polyps, primary tumor and metastatic lesions were biopsied and analyzed using transcriptomics (GEO GSE41258).[32,33] We calculated module perturbation scores of the 25 progression-associated modules discovered on TCGA's colon cancer cohort and compared the distributions of MPS in the different tissue types on this external cohort (Mann-Whitney test). We also sought to examine pathways relating to apoptosis and oxidative phosphorylation, previously discovered by Drier and colleagues to be correlated with disease progression in this colon cancer cohort using the pathway deregulation scores (PDS).[15,71] PDS utilizes normal samples as a reference to quantify the extent of deregulation of a pathway in individual samples. For both modules, with identical module definitions, we calculated their PDS (R package 'pathifier')[71] as well as their MPS and quantified the correlation coefficients between the two measures. While these specific modules did not meet our statistical thresholds for identifying modules associated with cancer stage in TCGA, we did find other significant progression-associated modules that shared substantial similarities with each of these modules (hypergeometric test p value $<10^{-5}$).

### Validating prognosis of molecular features

To validate the prognostic value of modules in independent cohorts, we performed tissue-matched comparisons between TCGA and independent collections from ICGC and ACRG (labeled external cohorts). For each PCM discovered in a TCGA cohort, we quantified the Z score for the mutual information value between the module's gene membership and every individual patient transcriptome in the

matched independent cohort and MPS values as described earlier. We assessed survival differences between patients with activated and repressed perturbations scores with similar number of patients in the two groups, as described for TCGA cohorts. The number of patients used in each group (n) was decided based on the size of the cohort: clear cell kidney cancer (ICGC-Europe; n = 25), liver cancer (ICGC-Riken; n = 25, 50, 100), ovarian cancer (ICGC-Australia; n = 25), pancreatic cancer (ICGC-Canada; n = 25, 50) and stomach cancer (ACRG; n = 25, 50, 100, 150), only including comparisons with at least twenty patients in each group. PCMs that resulted in survival differences with log rank test p value <0.05 or absolute $\log_2$(hazard ratio) > 0.4 were defined to be prognostic in the external cohort. Only PCMs that were prognostic in the external cohort in the same direction as the tissue-matched TCGA cohort (e.g., activated MPS specified worse prognosis in both cohorts) were considered to be consistently prognostic in the two cohorts. For every cohort, to assess the significance of this overlap, we assessed the prognostic ability of randomly sampled modules in the external cohorts (as described above). The same number of modules as there were PCMs in TCGA were randomly chosen from the set that was not prognostic in TCGA. From 100,000 repetitions, we quantified the statistical significance of finding a greater number of prognostic modules in the random set than in the set of PCMs for each cohort. For gene-level comparisons between TCGA and the external cohorts, genes were defined to be consistently prognostic in both cohorts based on the same p value and hazard ratio thresholds used for modules. We used area under the receiver operating characteristic curves (ROC) to evaluate the predictive performance of modules and genes on independent external cohorts. The positive sets used on AUC estimation for single-loci are defined based on genes whose expression was prognostic in tissue-matched cohorts in TCGA. Negative sets for genes were defined by randomizing each transcriptomic profile from the independent cohorts followed by standardizing the randomized gene expression profiles. On these gene expression values (n = 100 for each cohort), survival analyses were carried out as described earlier. For modules, PCMs discovered on tissue-matched TCGA cohorts were defined as positive sets. Negative set for modules (n = 100) were obtained by randomizing module perturbation scores for each sample in the independent cohorts followed by patient stratification and survival analysis as described above. These positive and negative sets were used to compute AUC. The procedure was repeated 10 times for each cohort and the distributions of AUC values for gene expression and PCMs were compared for each cohort. To compare the molecular and clinical features between cohorts in the two collections, we visualized the relative dissimilarities amongst the transcriptomic data using tSNE (R package 'Rtsne' 0.15).[72,73]

## Prognostic modules in patient sub-groups

We explicitly partitioned patients in each cohort based on significant clinical and patient features including histological type, tumor stage, age, sex, race as well as important cohort-specific clinical factors and prominent molecular features.

Additionally, we controlled for annotated molecular states (BCR/ABL status, FLT3 mutation status, PML-RARA status, IDH1 status) in AML; hormone receptor status (estrogen receptor (ER) and progesterone receptor (PR) statuses both negative or not), HER2 receptor status (negative or not), triple-negative status (ER, PR and HER2 receptor status all negative) in breast cancer; HPV status for cervical, esophageal and head and neck cancers; microsatellite instability status and side of tumor procurement (i.e., location) in colon cancer and Gleason score (high: 8 and above; medium/low: 7 and below) for prostate cancer.

Furthermore, we utilized prominent mutations and copy-number aberration patterns to partition patients in each cohort. For mutations, in each cohort, we selected only genes that harbored mutations in at least 5% of patients for which primary tumor transcriptomic data was also available. From this set, we chose at most 3 genes (excluding TTN) with the highest number of mutations per base pair of its coding sequence. These included (in order) NPM1, DNMT3A, NRAS in AML; TP53, PIK3CA, KDM6A in bladder cancer; TP53, PIK3CA, GATA3 in breast cancer; PIK3CA, HLA-A, PTEN in cervical cancer; KRAS, TP53, NRAS in colon cancer; TP53, CDKN2A, NTM in esophageal cancer; TP53, PTEN, EGFR in GBM; TP53, IDH1, PTEN in glioma; TP53, CDKN2A, PIK3CA in head and neck cancer; VHL, PBRM1, MUC4 in kidney cancer; TP53, CTNNB1, ALB in liver cancer; KRAS, TP53, STK11 in lung adenocarcinoma; TP53,CDKN2A, TPTE in lung squamous cell carcinoma; NRAS, RPS27, CDKN2A in melanoma; TP53, USH2A, CSMD3 in ovarian cancer; KRAS, TP53, CDKN2A in pancreatic cancer; HRAS, MLLT3, POTEC in paraganglioma; SPOP in prostate cancer; KRAS, TP53, OPCML in rectal cancer; TP53, RB1, NRXN1 in sarcoma; TP53, KRAS, GNAS in stomach cancer; TVP23C, KRAS, SEC22B in testicular cancer; GTF2I, HRAS in thymoma; BRAF, NRAS in thyroid cancer; PTEN, KRAS, TP53 in uterine endometrial cancer cohorts.

To select relevant copy-number aberrations within each cohort, we computed the rank-product for each gene, $RP(g)$, based on its rank (ascending order) by the absolute median copy-number aberration value across all samples in the cohort ($r_M^g$) and its rank by the SD of the aberration value in the same cohort ($r_\sigma^g$):

$$RP(g) = \sqrt{r_M^g \cdot r_\sigma^g}$$

Starting from 1000 genes with the highest rank-product scores, we filtered out genes that were not relevant to cancer based on annotations on the OncoKB[27] database. Next, we iteratively selected the gene with the highest rank-product score while removing genes whose copy-number aberration values were highly correlated with the selected gene (|correlation| > 0.5). We selected up to 3 genes with the highest rank-product that were least likely to have correlated copy-number aberration patterns within the cohort. These included XRCC2, PTPRS, TP53 in AML; CDKN2B, NKX3-1, UBR5 in bladder cancer; MYC, NKX3-1, MCL1 in breast cancer; PRKCI, CHEK1, YAP1 in cervical cancer; GNAS, ATP6V1B2 in colon cancer; CDKN2B, MYC, PRKCI in esophageal cancer; CDKN2B, FAS, EGFR in GBM and glioma; CDKN2B, SOX2, PDCD1LG2 in head and neck cancer; MLH1, MET, FLT4 in

kidney cancer; ATP6C1B2, MYC, MCL1 in liver cancer; TERT, MYC, SETDB1 in lung adenocarcinoma; DCUN1D1, CDKN2B, WWTR1 in lung squamous cell carcinoma; CDKN2B, CD272, MOB3B in melanoma; MYC, NKX3-1, TP63 in ovarian cancer; CDKN2B, SMAD4, MAP2K4 in pancreatic cancer; NRAS, GPS2 in paraganglioma; ESCO2, WHSC1L1, SETDB2 in prostate cancer; FLT1, PTPRT, SMAD4 in rectal cancer; CYSLTR2, CDKN2B, BRCA2 in sarcoma; MYC, GNAS, DUSP4 in stomach cancer; ETNK1, CHEK1, SOX17 in testicular cancer; NF2, IRF8, JARID2 in thymoma; XBP1, CDC73, TRAF2 in thyroid cancer; RIT1, NKX3-1, MYC in uterine endometrial cancer cohorts. To partition patient samples based on the copy-number aberration values of each selected gene, samples were categorized as 'deep amplified' if the copy-number aberration values are in the top 25% of positive copy-number values, 'amplified' for rest of samples with positive copy-number aberration values, 'deep deleted' if the values are in the bottom 25% of non-zero negative copy-number values, 'deleted' for rest of the samples with negative copy-number values.

For every clinical and molecular attribute detailed above (e.g., histological types in breast cancer; HPV status in cervical cancer; pathologic stage in stomach cancer etc.), we partitioned patients in that cohort within these sub-groups based on values of the attribute (e.g., Infiltrating Ductal Carcinoma or Infiltrating Lobular Carcinoma in breast cancer; stage I/IA/IB or stage II/IIA/IIB or stage III/IIIA/IIIB/IIIC or stage IV in stomach cancer; TP53 wild-type or TP53 mutant pancreatic cancer; MYC deep amplified or MYC amplified or MYC deleted or MYC deep deleted breast cancer). Within each of these sub-cohorts, we identified modules that were clinically prognostic in 3-fold cross-validation mode with the same training/test data partitions as with the full cohort. We performed these analyses only if the size of the sub-cohort included more than 20 samples and, at least 10 training samples in the MPS$^+$ and MPS$^-$ groups for each of the 3-folds of cross-validation and, 10 test samples labeled MPS$^+$ and MPS$^-$ after boundary conditions were propagated from the training set. We defined a module to be prognostic in a given sub-cohort when survival differences were significant in each of the 3 training splits (log-ratio rank test FDR <0.2) and in the label-propagated test set (log-ratio rank test FDR <0.1) and in the full cohort (without training/test split) with the same fixed number of samples (log-ratio rank test FDR <0.05). More relaxed thresholds reflect the smaller sizes of the sub-cohorts.

### Biological processes enriched in modules
We carried out Gene Ontology enrichment analyses to identify processes underlying genes comprising each module to provide additional functional context (R package 'topGO' 2.22.0[74]).

### Random survival forest models trained on PCMs
We evaluated the combined prognostic ability of modules that were individually identified to be prognostic within a cancer cohort. We used random survival forests (implemented in R package 'randomForestSRC' 2.7.0[75]), a non-parametric ensemble learning random forest approach applied to right-censored survival data.[54,76] Approaches based on random forests are powerful since they incorporate patterns of correlations and dependencies between module perturbations, potentially capturing interactions between the modules. We used 10-fold cross-validation with 1000 trees with the average number of unique data points in a terminal node set to 15 and the number of variables randomly selected for splitting a node set to be the ceiling of the natural logarithm of number of features (ceiling function defined earlier) with log rank split as the splitting rule. For each training-test split in the 10-fold cross-validation, we collated the predictions from the forest (i.e., estimated risk) of patients in the test sets to compile the full set of predictions of all samples. Finally, patients were sorted based on risk predicted by the random survival forest model and categorized into high and low risk groups to quantify survival differences between them. Comparisons between survival stratifications from module perturbation scores of individual PCMs and random survival forest models (number of predicted high- and low-risk patients) were made on similarly sized group of patients.

In order to test the cross-cohort predictive value of modules, we first trained a random survival forest model for a cancer cohort in the TCGA collection only using PCMs identified in that cohort. We then used this model to make survival predictions in every other cohort in the TCGA. The top and bottom 40% of patients (by estimated risk) in the predicted cohort were used for survival comparisons. Finally, we also used these models to make predictions in tissue-matched comparisons on the external cohorts by clustering the ensemble survival probabilities at event times as predicted by the random survival forest models.

### Prognosis of modules and clinical features
We used random survival forest models to assess and compare the abilities of different types of features to predict patient overall and progression-free interval survival in each cohort. We separately considered survival predictions of models trained on module perturbation scores of PCMs in each cohort, standard clinical factors, SNVs, CNAs and inferred immune cell-type frequencies. We excluded the pan-kidney cancer cohort from these analyses, given that this cohort is not considered to be a single disease.[77] For each individual feature category, we trained random survival forest models with 10-fold cross-validation, repeating the process 10 times. To assess the predictive ability of each model, we compared the survival differences of high-risk and low-risk patients, categorized based on the model's predictions. To assess the relative prognostic strength of the different feature categories, we compared the distributions of the absolute $\log_2$-transformed hazard ratios (n = 10; one-sided Mann-Whitney test). We also assessed if module perturbation scores improved the prognostic abilities of standard clinical factors, SNVs, CNAs or inferred immune cell-type frequencies by comparing survival predictions of random survival forest models trained on each of these individual feature categories alone and in combination with module perturbation scores. In each cohort, the same number of predicted high- and low-risk patients were chosen for survival comparisons for all the models. For survival comparisons, number of patients in the high and low risk groups were chosen to be:

$$min\left(\left|\left|\left\lceil\frac{f \times N}{50}\right\rceil \times 25 - [25\ 50\ 100\ 150\ 200\ 250\ 300\ 350\ 400\ 450\ 500]\right|\right|\right),$$

where $N$ is number of patients in the cohort and $f$ is a fraction between 0 and 1. Thus, $fxN$ is the number of samples used for survival comparisons of patients stratified based on risk predictions from the different random survival forest models. Different choices for $f$ (0.4, 0.5, 0.666, 1) had little qualitative impact on the comparisons between the random survival forest models and for all comparisons we chose $f$ to be 0.666. In the prostate cancer cohort, for survival comparisons on predictions from models trained on module perturbation scores of PCMs, standard clinical factors and SNVs (or CNAs), the Cox proportional hazards model provided invalid Wald statistic because of singularities that arise during coefficient estimation (manual PAGE for 'survival' package in R[69]). For these comparisons, $f$ was chosen to be 1.

### Standard clinical factors
We included histological type, tumor stage (both AJCC and discretized) and age groups (patient's age <30, 30 < age <60, age >60) for all cohorts. We also included additional features in AML-molecular test status; breast cancer-estrogen receptor status, progesterone receptor status, HER2 receptor status, triple-negative status; cervical cancer- HPV status; colon cancer-microsatellite instability status, side of tumor procurement; head & neck cancer- HPV status and prostate cancer- Gleason score, Prostate-specific antigen value.

### SNVs
We selected prominently mutated genes as described earlier. We included up to 50 genes that were mutated in more than 30 samples. These analyses were only performed in cohorts that had both transcriptomic data as well mutation data in more than 100 primary tumor samples.

### CNAs
In each cohort, we computed the rank-product for each gene, $RP(g)$ (defined earlier). As random model forest features, we chose 10% of genes with the highest rank-product scores and iteratively selected the genes with the highest rank-product score while removing genes whose copy-number aberration values were highly correlated with the selected gene (|correlation| > 0.5). Thus, we included genes with the highest rank-product that were least likely to have correlated copy-number aberration patterns within the cohort. We categorized samples based on the copy-number aberration values of each gene as 'deep amplified' if the values are in the top 33% of positive copy-number values, 'amplified' for rest of samples with positive copy-number aberration values, 'deep deleted' of the values are in the bottom 33% of non-zero, negative copy-number values, 'deleted' for rest of the samples with negative copy-number values.

### Immune cell-type abundances
We utilized the inferred relative frequencies of 22 immune cell-types in TCGA's primary tumor transcriptome data.[58] These frequencies were estimated using CIBERSORT (cell-type identification by estimating relative subsets of RNA transcripts) which uses pure-population reference profiles of immune cell-types to deconvolve bulk transcriptome data to estimate the relative proportions of the different cell-types. The data were downloaded from GDC (Genomics Data Commons; https://gdc.cancer.gov/about-data/publications/panimmune).

## QUANTIFICATION AND STATISTICAL ANALYSIS

Analyses were carried out using the R statistical computing environment[63] (versions 3.2.3, 3.6.3). The code used for the analyses can be found at https://github.com/balaji-srinivasan-santhanam/pcm. Details of specific functions and libraries are provided in the methods sections. For statistical tests, details of the specific tests used and parameters are specified at appropriate sections of the text including in the methods, results and figure legends sections. Survival comparisons were quantified using log rank test and Cox proportional hazards model. Between-group comparisons were made using Mann-Whitney U test (Wilcoxon rank-sum test). Contingency table tests were performed using hypergeometric test or Fisher exact test.

## ADDITIONAL RESOURCES

Data are available as Data Sheets. Results have also been made available at https://tavazoielab.c2b2.columbia.edu/PCMs/landing_page.html

