## [Document S2. Transparent Peer Review record for Santhanam et al · Cell Genomics]

Systematic assessment of prognostic molecular features across cancers

Author list

Balaji Santhanam, Panos Oikonomou, Saeed Tavazoie

Summary**Initial submission:** Received : March 28th 2022

Scientific editor: Rosalind Mott and Judith Nicholson

First round of review: Number of reviewers: 3
Revision invited : August 23rd 2022
Revision received : September 29th 2022**Second round of review:** Number of reviewers: 2
Accepted : 12th January 2023**Data freely available:** Yes**Code freely available:** Yes

This transparent peer review record is not systematically proofread, type-set, or edited. Special characters, formatting, and equations may fail to render properly. Standard procedural text within the editor's letters has been deleted for the sake of brevity, but all official correspondence specific to the manuscript has been preserved.

Referees' reports, first round of review

Reviewer 1

This is a very thorough serious study, and I believe it should be accepted for publication after some revision which in my opinion will improve the paper.

1. General critique:

1.1 No comparison with any previous work

The paper introduces a new measure of Module Perturbation, based on mutual entropy of module membership of a gene and its expression levels in a cohort. A Module Perturbation Score (MPS) is calculated from expression data of a particular cohort/cancer type. It is derived for every different module and each individual patient/sample.

Similar measures have been introduced in prior work - the closest that comes to mind is the Pathway Deregulation Score of Pathifier (Drier et al 2013) which seems to be the most widely used one out of a plethora of methods developed during the last decade (not counting the more standard enrichment based scores).

Unfortunately, the practice of proposing a new method without testing it against widely used existing ones is fairly common in the field, but I find it unacceptable. In the present case, such a comparison is scientifically important, since MPS assumes that the main effect of deregulation of a pathway is a more or less uniform shift of expression of its constituent genes. This apparently does not always hold; (A) some pathways are deregulated by increase of expression of some genes and decrease of others; (B) expression sometimes varies with progression in a non-monotonic manner, and the trajectory in expression space is non-linear and non-monotonic. Deregulation of pathways that exhibit A or B could be missed by the MPS.

I also miss validation of some direct biological significance of the proposed MPS. The most direct test is to check correlation of MPS with progression, rather than presenting evidence for prognostic capacity, which is a much less understood and undisputed property.

To respond to these weaknesses, the authors should take one or two datasets that were studied using Pathifier, calculate for each sample the PDS and the MPS for identical pathway modules, and investigate the correlation of the two scores over the samples of a cohort, and of the scores with progression. I recommend

colon cancer data, for which normal tissue, polyps, primary tumors and metastatic samples provide a clear indication of progression. Another possibility is breast cancer, in particular - the very extensive METABRIC dataset. See Livshits et al Mol Onc 9 (2015) 1471.

1.2 The Introduction states or implies that no previous pan-cancer pathway based analysis has been performed. This is incorrect. I tend to agree that the current study is probably more thorough in comparisons with single-gene based prognostic predictors (more about this later) than most other studies, but here are a few I found (this is the result of a superficial recollection, not a thorough literature search):

Rau et al Biostatistics (2022) 23, 2, pp. 362-379 TCGA breast and lung

Li et al Bioinformatics, 2022, 1-8 <https://doi.org/10.1093/bioinformatics/btac122>
10 cancers TCGA

Fang et al Genes 2020, 11, 1281 TCCGA 16 cancers

Bao et al IEEE Access 2020 Digital Object Identifier 10.1109/ACCESS.2020.3010796
28 cancers

Zheng et al. BMC Bioinformatics (2020) 21:76 <https://doi.org/10.1186/s12859-020-3423-z> TCGA more than 20 cancers, prognosis

Claims of being the first to do something should be made with a lot of care.

1.3 Comparison of module-based versus single gene-based prognosis:

In some sense the way these comparisons are made is unfair; consider for example the prognostic value of a module built around a particular driver gene, being higher than the prognostic value of expression of the driver alone.

Clearly, using expression levels of say 30 genes that constitute the module (and include the driver) contains more information than the driver alone, so the module-based score should be more informative about survival.

The more "fair competition" is between prognostic classifiers constructed out of 50 - 100 genes, that are NOT associated with a particular pathway - but were selected using some feature selection method, usually by ranking all genes on the basis of, say, correlation of expression with survival. The number of "prognostic genes" should be close to that of the genes of the modules considered. Before looking at prognostic value, one should note that ranking of single genes was shown to be highly unstable, with independent cohorts (of the same disease) yielding widely varying "top-ranked prognostic gene lists". On the other hand, the prognostic ability of modules was claimed (e.g. Cancer Res 74(17) September 1, 2014) to be much more robust than that of individual genes. This claim is apparently supported also by the present study; the prognostic value of a module,

discovered for the TCGA cohort, is reproduced in independent cohorts (page 10, bottom).

2. Particular comments:

Fig1 a: the caption mentions overall survival and progression free interval. Only one figure is shown; is it the average of the two from S1?? Or of the two combined? Fig S1 does have two separate figures for the same, but that of 1a is different from both.

Fig 1c: the side plot for patient B (the blue blob on the right) should be raised above the grey horizontal line *it corresponds to genes in module, down-regulated).

Fig 1d: I did not understand - what does this figure is supposed to show? that for the P53-associated module, the samples with $MPS > 0$ had a higher fraction with mutant p53 than the samples with $MPS < 0$? Is it not simpler to state this in the text than by a confusing figure?

*** mutation status of p53 - not all mutations cause loss of function, hence I am not surprised that any reasonable measure of P53 activity (such as embodied by module activity) has a better association with survival than mutation status.

Fig S4: for each cohort/cancer there are modules whose MPS is positively correlated with expression (or CNA) and other modules with negative correlation. Red triangles denote statistically significant correlations and grey circles - non-significant ones. I recommend using the shape for significance and the color to identify tumor suppressors (blue) and oncogenes (red), with which each of the 67 modules is associated.

Fig 1f: This is, as I understand, a comparison between (a) the level of significance of survival stratification based on the MPS of a module, associated with a cancer driver gene, and (b) that of expression (or CNA or mutation) of that same cancer driver gene. The fact that one looks at that gene (and not say the "best" gene of the module) should be stated (it appears in the legend of subsequent figures....). Why 19 cohorts and not 25?

Figure S9b; how were the 1978 modules selected? Are these the union of all PCMs? It would help to present in a heatmap the overlap matrix between the constituent groups of genes - do modules that cluster together share a higher than expected number of genes? I noticed that perhaps Fig 3e answers to some extent this question for a particular set of modules.

Figure S9c; the fact that normal samples and tumor samples exhibit different expression patterns has been known for 20 years. The potentially new and interesting feature of this figure is the fact that differentiating modules are

different for different cancers, but this is obscured by this figure. All that one can see is that the number of significantly differentiating modules differs from cancer to cancer. Perhaps a matrix (or heatmap) whose rows and columns represent the different cancers and the entry representing the number of shared PCMs of the two cancers - will help, but some discussion of who are the shared and the cancer-specific modules is also interesting.

Using discovered gene regulatory motifs as a means to assemble previously unknown regulatory modules is an interesting new idea, and the reported results on the prognostic value of some of these new modules is interesting. I recommend in all figures (in the paper and the Supplementary) that describe prognostic or other features of a particular regulator-centered module, to place in parentheses the number of genes that belong to the module. This may be hidden somewhere in an online table, but I did not find it, and it is of interest to give the reader an idea of the number of genes that were identified as members of a PCM. For example, in Fig. 3a-d and in Fig S12 - how many genes belong to the RUNX2-based module?

I was impressed by the attempt to find PCMs that work for a particular subtype of a cancer, e.g. triple negative breast cancer. Personally I believe that findings of highest clinical relevance will most likely be limited to particular subtypes. I also liked the attempt to use multiple PCMs to improve on the prognostic capacity - it is a natural path to take, rather than relying on one PCM at a time. Again, this was done before in both supervised and unsupervised manner, but it is nice to see that effort was made here to perform such machine learning tasks for multi-module analysis.

Reviewer 2

Excellent work done, and congratulations on that.

One minor request that when it will be published, please make sure your produced/reproduced results with open source code and a tutorial guiding to how to practice your research will be a great support to the community.

Reviewer 3

This paper shows extensive data that changes in mRNA expression by multiple genes has superior prognostic value to single mutations or other single gene "indices" such as copy number variation. Intuitively, this is expected as changes in

RNA expression amongst of groups of genes would be expected to reflect cell biology and balance of cell population regulation better than presence of "single" abnormalities.

The discussion should reflect on certain limitations inherent in the cases sampled.

1. The cases were likely sampled at time of initial surgery; prognosis as determined by overall survival, was measured variably from months to several years later.

This implies that cases with poor prognosis likely had gene expression features that not only facilitated spread but also made the cancers refractory to varied tumor specific chemotherapy, endocrine therapy radiotherapy and immunotherapy.

2. Under normal circumstances, gene expression and presumable modules of gene expression can change rapidly with changes in the cell environment. The various therapies would have changed the number and kind of residual cancer cells as well as dramatically affecting the cell environment within the tumor. Yet , gene expression modules at initial case presentation was a good prognostic determinant. Why?

3 The prognostic value of Module Perturbation Scores varied greatly by Cancer tissue type, being best for Breast and Prostate and of lesser value for other cancers. Why?

4. It is likely that cases sampled were obtained from Tertiary Cancer Centres. This may have biased cases to larger or more advanced cancers. This can be checked by comparing cancer stage with prevalence of each stage in general cancer statistics. The cases may have been more biologically aggressive than average. This could have been assessed by tumor histologic grade that while imperfect, is a measure of aggressiveness.

Authors' response to the first round of review

We would first like to thank the reviewers for their thoughtful comments, constructive feedback and useful suggestions. We believe that our manuscript has immensely benefitted from them. Broadly speaking, the reviewers recommended

that we: (i) further contextualize our work with prior literature, (ii) provide additional biological significance and conceptual clarity of module perturbation scores and, (iii) highlight limitations of the proposed approach and the datasets utilized. There were also several minor suggestions to clarify and/or simplify our narrative. In this current version, we have addressed all of the comments of all three reviewers. Our revision included additional analyses, for which we have added new supplemental figures and sub-panels. The changes are reflected in the appropriate sections in the text and have been marked. We believe that our revised manuscript provides better clarity and further substantiates our original analyses and observations. Below, we have provided a point-by-point response to the reviewer comments by including specific changes made in the text as well as key figures to provide context.

Reviewer #1

This is a very thorough serious study, and I believe it should be accepted for publication after some revision which in my opinion will improve the paper. 1. General critique: 1.1 No comparison with any previous work The paper introduces a new measure of Module Perturbation, based on mutual entropy of module membership of a gene and its expression levels in a cohort. A Module Perturbation Score (MPS) is calculated from expression data of a particular cohort/cancer type. It is derived for every different module and each individual patient/sample. Similar measures have been introduced in prior work - the closest that comes to mind is the Pathway Deregulation Score of Pathifier (Drier et al 2013) which seems to be the most widely used one out of a plethora of methods developed during the last decade (not counting the more standard enrichment based scores). Unfortunately, the practice of proposing a new method without testing it against widely used existing ones is fairly common in the field, but I find it unacceptable. In the present case, such a comparison is scientifically important, since MPS assumes that the main effect of deregulation of a pathway is a more or less uniform shift of expression of its constituent genes. This apparently does not always hold; (A) some pathways are deregulated by increase of expression of some genes and decrease of others; (B) expression sometimes varies with Response to Reviewers progression in a non-monotonic manner, and the trajectory in expression space is non-linear and non-monotonic. Deregulation of pathways that exhibit A or B could be missed by the MPS.

We thank the reviewer for their insightful comments. Our computational framework is based on the concept of mutual information which captures

monotonic and non-monotonic dependencies between observations. While, for simplicity of interpretation, we refer to coordinated shifts in expression within a pathway or module, we fully agree with the reviewer that this may not always be the case. To address this issue, we carried out simulations in which we computationally constructed modules of varying sizes by randomly sampling genes with both high and low expression values (figure shown below). We computed the perturbation scores for these synthetic modules. As may be expected, we found that the absolute value of module perturbation scores was largest for modules that only comprise genes with either low expression (top row in heatmap) or high expression (bottom row in heatmap), regardless of module size (indicated in column headers). However, the magnitude of module perturbation scores is still highly statistically significant ($p < 10^{-88}$) for mixed expression modules, though it is lowest for modules that contain an equal mix of genes with high and low expression values (middle row in heatmap). Our results suggest that constructing fine-grained modules comprising genes that show coherent changes in their expression patterns will certainly boost MPS values thereby improving sensitivity for downstream analyses. To point this out, we have now made note that genes within a module can exhibit both increase and decrease in their expression in the Discussion section and by including a new subpanel in Supplemental Figure S5. The subpanel (Fig S5c) shows three example modules that constitute genes with both high and low expression in their respective cohorts. Despite having genes with mixed expression patterns, we find that the module perturbation scores are statistically significant and furthermore, can discriminate patients based on their survival (survival plots in Fig 1d, Fig S5a-b; right panel). We have highlighted these points in the Discussion section and in the figure legends. In the Discussion, we state

“Our results suggest that module perturbations capture at least some of the molecular complexity underlying cancer, reinforcing the advantages of utilizing gene groups over single-locus observations (Domany, 2014). The information-theoretic basis of our computational framework enables the quantification of monotonic and non-monotonic dependencies between module membership and patient transcriptomes (Cover and Thomas, 2006; Goodarzi et al., 2009). While MPS can reflect gene expression perturbations within a module including non-uniform changes (examples shown in Fig S5c), coarse and non-specific module definitions may affect the sensitivity of our approach. Thus, building fine-grained modules comprising genes with coherent changes in their expression, could not only boost MPS values, but also improve downstream analyses. Additionally,

multivariate phenotypic attributes such as survival may exhibit non-monotonic dependencies with the patient's molecular characteristics. Supplementing our current framework with state-of-the-art statistical and machine-learning models may be essential to discovering these relationships."

Perturbation scores of a simulated module comprising genes with high and low expression. (a) Distribution of standardized expression values for a patient transcriptome randomly selected from a cohort of 342 colon cancer patients (GEO accession GSE41258; patient ID A5135A) (Drier et al., 2013; Martin et al., 2018; Sheffer et al., 2009). These values have been discretized and binned into 10 equally populated bins (bin numbers in yellow). Expression range for the genes within each bin is indicated (red box; values shown). (b) Each row corresponds to a module constructed by randomly sampling genes with low expression (bins 0-1) and high expression (bins 8-9). Fraction of genes with low and high expression that constitute the module is tabulated on the left and, the size of the module is indicated in column headers of the heatmap. Module perturbation scores are calculated using the discretized expression vector (panel a; 10 bins) and the binary module membership vector. For each of the modules, their module perturbation scores are indicated (heatmap; right panel) and colors correspond to the sign of the perturbation scores. These results were largely reproducible for other choices of transcriptomes.

Figure S5: Perturbations in modules associated with KRAS and PTEN provide better patient survival stratification than measurements on individual genes. (a-b) Patients stratified based on mutation status, copy-number aberrations and expression changes in (a) KRAS in lung adenocarcinoma (LUAD) and, (b) PTEN in stomach cancer (STAD). Also shown are patient stratifications based on perturbations in module comprising (a) genes up-regulated in lung cancer cells over-expression KRAS (MSigDB (Liberzon et al., 2015a) M12860) and, (b) genes up-regulated upon knockdown of PTEN (MSigDB (Liberzon et al., 2015a) M2787). (c) Violin plots show distributions of the mean expression of genes that make up the modules corresponding to repressed targets of TP53 (MSigDB M2698; 198 genes) in pancreatic cancer (left panel), activated targets of KRAS (MSigDB M12860; 139 genes) in lung adenocarcinoma (middle panel) and repressed targets of PTEN (MSigDB M2787; 186 genes) in stomach cancer cohort (right

panel). Patients have been stratified into groups with positive (red) and negative (blue) perturbation scores for these modules. Genes that constitute the modules are shown (black dots) and their mean expression values in the MPS positive and negative groups are linked (black lines). Slopes of the lines indicate whether genes are up- (negative slope) or down-regulated (positive slope) in the MPS positive compared to the MPS negative patient groups. Each module shown comprises of genes with both high and low average expression, while its perturbation scores is statistically significant and it significantly discriminates patients based on survival. Stratification of patients based on the perturbation scores of the three modules shown in this sub-panel are plotted in Fig 1d, Fig S5a (right panel) and Fig S5b (right panel), respectively

I also miss validation of some direct biological significance of the proposed MPS. The most direct test is to check correlation of MPS with progression, rather than presenting evidence for prognostic capacity, which is a much less understood and undisputed property. To respond to these weaknesses, the authors should take one or two datasets that were studied using Pathifier, calculate for each sample the PDS and the MPS for identical pathway modules, and investigate the correlation of the two scores over the samples of a cohort, and of the scores with progression. I recommend colon cancer data, for which normal tissue, polyps, primary tumors and metastatic samples provide a clear indication of progression. Another possibility is breast cancer, in particular - the very extensive METABRIC dataset. See Livshits et al Mol Onc 9 (2015) 1471.

We thank the reviewer for their constructive feedback and suggestions. Inspired by the reviewer's comments, we have now carried out additional analyses to systematically discover modules whose perturbation scores robustly associated with disease progression. For these analyses, we used cancer stage as the phenotype of interest- a proxy for disease progression, on primary tumors from 15 cancers in TCGA (Methods). For each module, we tested for preferential enrichment/depletion patterns of patients labeled MPS+ and MPS- across the four cancer stages in each cancer. Overall, our analyses identified 3221 modules in 12 cancers whose perturbation scores were significantly associated with cancer stage. We have presented the results of these analyses as a new Data Sheet (Data Sheet 3) and a new figure (Fig S10). A summary of these results is presented in Supplemental Figure S10a. To validate our results, as suggested by the reviewer, we utilized a cohort consisting of normal, polyps, primary tumor and metastatic samples obtained from 342 patients with colonic neoplasms (GEO accession

GSE41258) (Drier et al., 2013; Martin et al., 2018; Sheffer et al., 2009). Drier and colleagues identified pathways relating to apoptosis (115 genes) and oxidative phosphorylation (162 genes) to be significantly correlated with disease progression in this cohort based on their pathway deregulation scores (PDS) (Drier et al., 2013). While we observed significant correlation between PDS and MPS of these pathway modules (Fig S10b-c), these specific pathways did not pass our statistical thresholds. However, modules identified in our analyses shared substantial similarities with both apoptosis (GO term ‘mitochondrion’; 1980 genes) and oxidative phosphorylation (Reactome ‘Electron transport mediated ATP synthesis’; 151 genes) modules (hypergeometric p-values 10⁻¹¹ and 10⁻²³⁷, respectively) (Fig S10d-e). Finally, the perturbation scores of 88% of the modules discovered on TCGA’s colon cancer cohort significantly discriminated between cancerous tissues and colonic polyps and, between cancerous and non-cancerous tissues (Mann-Whitney test p-value < 0.05) (Data Sheet 3; tab ‘validation on GSE41258’). Some of these modules are highlighted in Fig S11. Taken together, these results suggest that our approach can discover modules significantly associated with disease progression in cancer, highlighting the versatility of our computational framework in examining phenotypes beyond patient survival. In the Results section, we state

“We also sought to test if module perturbation scores can capture disease progression. To this end, we utilized cancer stage as a proxy for disease progression, identifying 3221 modules in 12 cancers from TCGA (Methods; Data Sheet 3; Fig S10a). We found that the perturbation scores of modules relating to mitosis, extracellular matrix, RNA metabolism as well as transcriptional targets of polycomb group ring finger proteins BMI1 and PCGF2 were recurrently associated with disease progression in 3 or more cancers (Data Sheet 3). To test if our findings can generalize to different indicators of disease progression in independent patient groups, we utilized a cohort consisting of normal, polyps, primary tumor and metastatic samples obtained from 342 patients with colonic neoplasms (GEO accession GSE41258) (Drier et al., 2013; Martin et al., 2018; Sheffer et al., 2009). A majority of the modules discovered to be associated with colon cancer stage in TCGA were significantly associated with disease progression on this independent cohort. We identified modules with substantial similarities to apoptosis and oxidative phosphorylation pathways (hypergeometric p-value 10⁻¹¹ and 10⁻²³⁷, respectively), consistent with observations originally made by Drier and colleagues (Drier et al., 2013) (Fig S10b-e). We also discovered potentially novel modules relating to mitochondrial organization, protein localization and

targets of zinc finger transcription factor PATZ1 to be significantly associated with colon cancer progression (Fig S10-S11; Data Sheet 3). Clearly, further experimental validation is necessary to establish the *in vivo* functional roles of these modules in disease progression. Nonetheless, our results suggest that module perturbation scores can effectively capture biologically relevant phenotypes including disease progression.”

We have described the details of these analyses in the Methods section as “Cancer stage information was available for 15 cancers from TCGA. Primary tumors of stage I/IA/IB were grouped to be ‘stage I’, tumors of stage II/IIA/IIB/IIC to be ‘stage II’, tumors of stage III/IIIA/IIIB/IIC to be ‘stage III’ and tumors of stage IV/IVA/IVB/IVC were grouped to be ‘stage IV’. We modified our computational pipeline for discovering PCMs in order to identify modules whose perturbation scores are associated with disease progression, using cancer stage as a proxy for disease progression. After calculating perturbation scores on the full catalog of modules as described earlier, we chose a fixed number samples from MPS+ and MPS– groups (numbers specified for each cohort above). We then tested for preferential enrichment/depletion of either MPS group across the 4 discretized stages using Fisher’s exact test on the training and test datasets. We used the R function ‘fisher.test’ on the 4x2 contingency tables using default parameters with the flag ‘simulate.p.value’ set to TRUE to speed up calculations. We defined a module to be a progression-associated module in a cohort when enrichment/depletion patterns were significant (i) in each of the 3 training splits (Fisher’s exact test p -value < 0.05), (ii) in the label-propagated test set (Fisher’s exact test p -value < 0.01), and, (iii) in the full cohort (without training/test split) with the same fixed number of samples (Fisher’s exact test p -value < 0.01).”. Furthermore, we also state in the Methods,

“To validate modules associated cancer stage on an independent external cohort, we utilized a dataset consisting of 342 patients with colonic neoplasms from whom normal tissue, polyps, primary tumor and metastatic lesions were biopsied and analyzed using transcriptomics (GEO GSE41258) (Martin et al., 2018; Sheffer et al., 2009). We calculated module perturbation scores of the 25 progression-associated modules discovered on TCGA’s colon cancer cohort and compared the distributions of MPS in the different tissue types on this external cohort (Mann-Whitney test). We also sought to examine pathways relating to apoptosis and oxidative phosphorylation, previously discovered by Drier and colleagues to be correlated with disease progression in this colon cancer cohort using the pathway deregulation scores (PDS) (Drier, 2017; Drier et al., 2013). PDS utilizes normal

samples as a reference to quantify the extent of deregulation of a pathway in individual samples. For both modules, with identical module definitions, we calculated their PDS (R package 'pathifier') (Drier, 2017) as well as their MPS and quantified the correlation coefficients between the two measures. While these specific modules did not meet our statistical thresholds for identifying modules associated with cancer stage in TCGA, we did find other significant progression-associated modules that shared substantial similarities with each of these modules (hypergeometric test pvalue < 10⁻⁵).”

Figure S10. Module perturbation scores are correlated with cancer stage. (a) Number of modules that are significantly correlated with cancer stage in each module category (Y-axis; module categories indicated) across different TCGA cohorts (labels on top). (b-c) Scatterplots (left panel) show comparisons between Pathway Deregulation Scores (PDS; X-axis) and Module Perturbation Scores (MPS; Y-axis) for 342 samples obtained from patients with colonic neoplasms (GEO accession GSE41258) (Drier et al., 2013; Martin et al., 2018) for modules associated with (b) regulation of apoptosis (KEGG database; MSigDB M8492

(Liberzon et al., 2015b)) and (c) oxidative phosphorylation (KEGG database; MSigDB M19540 (Liberzon et al., 2015b)). Each dot corresponds to a patient and colors indicate tissue-types (key indicated) and the correlation coefficient between PDS and MPS of these two modules are shown. Significant correlation coefficients between the two measures suggest that that MPS of these modules also capture disease progression just like their PDS, as reported originally by Drier and colleagues (Drier, 2017; Drier et al., 2013). We compared the distributions of MPS between the biopsied tissue-types on the independent cohort and the statistics of these comparisons are shown as a heatmap (right panel; $-\log_{10}$ of two-sided Mann-Whitney test p-values shown). (d) Overlap between apoptosis pathway and module with highest similarity to it (based on hypergeometric test) is shown as a Venn diagram. Statistics of the overlap (hypergeometric test) and some of the prominent overlapping genes are indicated. The module associated with GO term mitochondrion (GO:0005739; 1980 genes) was one of the progression-associated modules discovered in TCGA's colon cancer cohort (panel (a)). (e) Overlap between oxidative phosphorylation pathway and module with highest similarity to it (based on hypergeometric test) is shown as a Venn diagram. Statistics of the overlap (hypergeometric test) and some of the prominent overlapping genes are indicated. The module associated with electron transport respiratory chain (Reactome R-HSA-163200; 151 genes) was one of the progression-associated modules discovered to be associated with colon cancer stage in TCGA (panel (a)).

Figure S11. Modules discovered on TCGA’s colon cancer cohort are correlated with disease progression on an independent colon cancer cohort. For progression-associated modules discovered on TCGA’s colon cancer cohort, the distributions of perturbation scores (modules indicated) in metastatic, primary tumor, polyps and normal tissues biopsied from patients with colonic neoplasms (GEO accession GSE41258) (Drier et al., 2013; Martin et al., 2018) are shown as violin plots (left panel). For each module, heatmap (right panel) shows statistics of the comparisons between MPS distributions in the biopsied tissue-types (right panel; two-sided Mann-Whitney test p-values shown).

1.2 The Introduction states or implies that no previous pan-cancer pathway based analysis has been performed. This is incorrect. I tend to agree that the current study is probably more thorough in comparisons with single-gene based prognostic predictors (more about this later) than most other studies, but here are a few I found (this is the result of a superficial recollection, not a thorough

literature search): Rau et al Biostatistics (2022) 23, 2, pp. 362-379 TCGA breast and lung Li et al Bioinformatics, 2022, 1-8
<https://doi.org/10.1093/bioinformatics/btac122> 10 cancers TCGA Fang et al Genes 2020, 11, 1281 TCCGA 16 cancers Bao et al IEEE Access 2020 Digital Object Identifier 10.1109/ACCESS.2020.3010796 28 cancers Zheng et al. BMC Bioinformatics (2020) 21:76 <https://doi.org/10.1186/s12859-020-3423-z> TCGA more than 20 cancers, prognosis Claims of being the first to do something should be made with a lot of care.

We agree with the reviewer that claims about being first should be made carefully. We have gone over the introduction section to ensure that claims we make are fair to published literature and is supported by evidence presented in this manuscript. Specifically, we have modified the line in the Introduction section from “These efforts have largely explored associations between perturbations in prominent cancer-relevant pathways with patient survival in only a handful of cancers.” to “These efforts have largely explored associations between perturbations in a limited set of prominent and curated cancer-relevant pathways with patient survival.”. We thank the reviewer for pointing us to an additional list of literature references; these have now been referenced appropriately in our manuscript.

1.3 Comparison of module-based versus single gene-based prognosis: In some sense the way these comparisons are made is unfair; consider for example the prognostic value of a module built around a particular driver gene, being higher than the prognostic value of expression of the driver alone. Clearly, using expression levels of say 30 genes that constitute the module (and include the driver) contains more information than the driver alone, so the module-based score should be more informative about survival. The more "fair competition" is between prognostic classifiers constructed out of 50 - 100 genes, that are NOT associated with a particular pathway - but were selected using some feature selection method, usually by ranking all genes on the basis of, say, correlation of expression with survival. The number of "prognostic genes" should be close to that of the genes of the modules considered. Before looking at prognostic value, one should note that ranking of single genes was shown to be highly unstable, with independent cohorts (of the same disease) yielding widely varying "top-ranked prognostic gene lists". On the other hand, the prognostic ability of modules was claimed (e.g. Cancer Res 74(17) September 1, 2014) to be much more robust than that of individual genes. This claim is apparently supported also by the present study; the prognostic value of a module, discovered for the TCGA

cohort, is reproduced in independent cohorts (page 10, bottom).

We thank the reviewer for this insightful point. We concur with the reviewer that the activity of groups of genes is likely to be more informative about patient survival than single genes. However, as the reviewer has noted above, to the best of our knowledge, this has not been systematically tested across several cancers. Based on results presented in our study, we also agree with the reviewer's comments on the limited predictive abilities of individual genes in independent cohorts. Our analyses show that (i) modules provide significantly better prognostic value than single-locus observations in TCGA cohorts (Fig 1e, Fig 2d-e), (ii) module perturbations are more predictive of patient survival in independent cohorts than expression changes in single genes (Fig 3f-g), (iii) in a majority of cancers, combinations of modules are more predictive of patient survival than combinations of single-genes (Fig 5e). Constructing modules that provide robust and strong patient survival stratification is indeed important and, utilizing groups of genes that are individually prognostic is an excellent first step. However, building computational tools to construct synthetic modules focused specifically on patient stratification are beyond the scope of this manuscript. Our work provides biological context and by utilizing modules comprising functionally related genes. PCMs offer insight on biological processes and regulatory factors relating to patient survival. In the future, we strongly believe this knowhow can even be leveraged for informing clinical decisions. We will incorporate the reviewer's suggestion in analyses that extend and build on our current work. We expect that a module comprising prognostic genes that are individually non-robust is likely to be robust. As the reviewer points out, this forms the basis of our work. However, a module constructed using unrelated but individually prognostic genes may not be readily interpretable in terms of its underlying biology. We have now stated in the Discussion *"While the modules utilized in this work were chosen for their ability to provide biological context, computationally constructing modules, such that their perturbation scores maximize survival differences can help generate robust modules highly predictive of survival."* We also want to thank the reviewer for pointing us to an excellent reference, which we have now included stating *"Our results suggest that module perturbations capture at least some of the molecular complexity underlying cancer, reinforcing the advantages of utilizing gene groups over single-locus observations (Domany, 2014)."*

. 2. Particular comments: Fig1 a: the caption mentions overall survival and progression free interval. Only one figure is shown; is it the average of the two from S1?? Or of the two combined? Fig S1 does have two separate figures for the

same, but that of 1a is different from both.

The figure shows the proportion of genes that are prognostic for both overall survival and progression-free interval survival. We have now clarified this point in the figure legends as follows: "Proportion of genes prognostic based on their copy-number aberrations (purple), mutation statuses (black) and expression changes (yellow) in each cohort (Y-axis) for both overall survival and progression-free interval survival."

Fig 1c: the side plot for patient B (the blue blob on the right) should be raised above the grey horizontal line *it corresponds to genes in module, down-regulated).

We thank the reviewer for raising an important point. This figure illustrates a scenario in which genes belonging to the module have lower relative-expression than other genes. We have clarified this point by including a scale for the Y-axis.

Fig 1d: I did not understand - what does this figure is supposed to show? that for the P53- associated module, the samples with $MPS > 0$ had a higher fraction with mutant p53 than the samples with $MPS < 0$? Is it not simpler to state this in the text than by a confusing figure? *** mutation status of p53 - not all mutations cause loss of function, hence I am not surprised that any reasonable measure of P53 activity (such as embodied by module activity) has a better association with survival than mutation status.

We thank the reviewer for their suggestion to simplify the narrative. We have now removed this subpanel.

Fig S4: for each cohort/cancer there are modules whose MPS is positively correlated with expression (or CNA) and other modules with negative correlation. Red triangles denote statistically significant correlations and grey circles - non-significant ones. I recommend using the shape for significance and the color to identify tumor suppressors (blue) and oncogenes (red), with which each of the 67 modules is associated.

We thank the reviewer for their suggestions. We have now modified figure S4 to highlight oncogenes and tumor-suppressor genes.

Fig 1f: This is, as I understand, a comparison between (a) the level of significance of survival stratification based on the MPS of a module, associated with a cancer driver gene, and (b) that of expression (or CNA or mutation) of that same cancer driver gene. The fact that one looks at that gene (and not say the "best" gene of the module) should be stated (it appears in the legend of subsequent figures....).

We have now clarified that the figure refers to comparisons between modules

associated with cancer-drivers and genes encoding the cancer-drivers themselves (new Fig 1e). Relevant text in the figure legends has been modified as follows: “The log₂-ratio of the absolute standardized significance of modules associated with cancer-drivers and measurements on genes encoding these cancer-drivers (in rows) are visualized in 19 cancers from TCGA.”. We have also made changes in the main text that state: “Remarkably, for ~90% of the cancer-drivers (Chakravarty et al., 2017), patient survival stratifications based on module perturbations were superior to stratifications based on measurements at the corresponding individual loci across the cancers tested (Fig 1e; Fig S5-S8).”

Why 19 cohorts and not 25?

For visualization purposes, the 19 cancers in which ~15% of either the modules or single-gene measurements associated with cancer-drivers had prognostic capacity are shown. The full set of cancers in which at least one cancer-driver was prognostic are shown in Fig S7 and Fig S8 (overall survival and progression-free interval survival, respectively).

Figure S9b; how were the 1978 modules selected? Are these the union of all PCMs? It would help to present in a heatmap the overlap matrix between the constituent groups of genes - do modules that cluster together share a higher than expected number of genes? I noticed that perhaps Fig 3e answers to some extent this question for a particular set of modules.

We have now explicitly stated the specific computational steps to select modules in the Methods section. Specifically, we state “*We selected cohort-specific modules such that they have high mean absolute module perturbation scores (top 50%) as well as low variance (bottom 40%) within individual cohorts while treating MPS+ and MPS- scores separately. We restricted our analyses to previously defined modules across 25 TCGA cancer cohorts, identifying 1978 modules with cohort-specific perturbation patterns.*”. We agree with the reviewer’s suggestion that pairwise overlaps between these modules could provide additional information. We have added a subpanel that shows the extent of overlap between modules and the cohorts they were identified in (new Fig S9c).

Figure S9c; the fact that normal samples and tumor samples exhibit different expression patterns has been known for 20 years. The potentially new and interesting feature of this figure is the fact that differentiating modules are different for different cancers, but this is obscured by this figure. All that one can see is that the number of significantly differentiating modules differs from cancer to cancer. Perhaps a matrix (or heatmap) whose rows and columns represent the different cancers and the entry representing the number of shared PCMs of the

two cancers - will help, but some discussion of who are the shared and the cancer-specific modules is also interesting.

In addition to the heatmaps showing modules dysregulated in matched tumor-normal comparisons (new Fig S9d), we have now added a new subpanel that visualizes the numbers of these modules shared between cancers (new Fig S9e). We have now provided lists of these modules discovered in a new Data Sheet (Data Sheet 3).

Using discovered gene regulatory motifs as a means to assemble previously unknown regulatory modules is an interesting new idea, and the reported results on the prognostic value of some of these new modules is interesting. I recommend in all figures (in the paper and the Supplementary) that describe prognostic or other features of a particular regulator-centered module, to place in parentheses the number of genes that belong to the module. This may be hidden somewhere in an online table, but I did not find it, and it is of interest to give the reader an idea of the number of genes that were identified as members of a PCM. For example, in Fig. 3a-d and in Fig S12 - how many genes belong to the RUNX2-based module?

We thank the reviewer for their feedback. We have now explicitly stated the number of genes that belong to a module in the figure legends.

I was impressed by the attempt to find PCMs that work for a particular subtype of a cancer, e.g. triple negative breast cancer. Personally I believe that findings of highest clinical relevance will most likely be limited to particular subtypes. I also liked the attempt to use multiple PCMs to improve on the prognostic capacity - it is a natural path to take, rather than relying on one PCM at a time. Again, this was done before in both supervised and unsupervised manner, but it is nice to see that effort was made here to perform such machine learning tasks for multimodule analysis.

We thank the reviewer for their constructive feedback and comments

Reviewer 2

Excellent work done, and congratulations on that. One minor request that when it will be published, please make sure your produced/reproduced results with open source code and a tutorial guiding to how to practice your research will be a great support to the community.

We thank the reviewer for their positive comments and encouragement. We have deposited code to our GitHub page and provided a web portal for the community

to access our results. Links for both are included in the manuscript.

Reviewer 3

This paper shows extensive data that changes in mRNA expression by multiple genes has superior prognostic value to single mutations or other single gene "indices" such as copy number variation. Intuitively, this is expected as changes in RNA expression amongst groups of genes would be expected to reflect cell biology and balance of cell population regulation better than presence of "single" abnormalities. The discussion should reflect on certain limitations inherent in the cases sampled.

We thank the reviewer for their comments. We have now added text on the limitations of the work presented here in relevant sections of the manuscript, including in the Discussion.

1-The cases were likely sampled at time of initial surgery; prognosis as determined by overall survival, was measured variably from months to several years later. This implies that cases with poor prognosis likely had gene expression features that not only facilitated spread but also made the cancers refractory to varied tumor specific chemotherapy, endocrine therapy radiotherapy and immunotherapy.

We thank the reviewer for their feedback. It highlights an important caveat in interpreting the prognostic potential of module perturbation scores. Specifically, based on the information available, we are not able to ascertain the degree to which the prognostic potential of module perturbations captures inherently aggressive disease as opposed to reflecting therapyresistant disease states. We expect that future inclusion of detailed therapeutic trajectories for cancer cohorts will enable us to associate prognostic effects of module perturbations to either innate disease aggressiveness or to relative efficacy of specific therapeutics modalities. We have now explicitly stated this caveat in the Discussion section as *"Since information on treatments and the efficacy of these treatments is generally not available, the degree to which the prognostic potential of module perturbation scores captures inherently aggressive disease as opposed to reflecting therapy-resistant states that facilitate spread, cannot be inferred from our results."*

2 Under normal circumstances, gene expression and presumable modules of gene expression can change rapidly with changes in the cell environment. The various therapies would have changed the number and kind of residual cancer cells as well as dramatically affecting the cell environment within the tumor. Yet , gene

expression modules at initial case presentation was a good prognostic determinant. Why?

We thank the reviewer for raising this insightful point. We concur with the reviewer that exposure to therapeutics can change the tumor's molecular compositions as well as its microenvironment. The remarkable successes of targeted genomic profiling of tumors biopsied around the time of initial diagnosis, to prioritize therapeutic choices for patients (Shaw and Maitra, 2019) motivated us to explore biomarkers beyond genomic features. Indeed, transcriptome profiling of tumors at the time of initial biopsy only provides steady-state measurements of gene expression. Yet, it is quite remarkable that module perturbations calculated from baseline steady state transcriptomes carry substantial information on patient survival outcomes. We postulate that MPS captures the collective effects of genomic and epigenomic perturbations to robustly reflect cancer cellular states. We also agree with the reviewer that temporal sampling of the molecular changes in tumors as they respond to treatments would provide increased resolution on the effect of treatments on tumors. We have no doubt that development of less invasive methodologies for longitudinal sampling of patient tumor cells as well as effective in vitro and/or ex vivo models that reproducibly mimic effects in human patients, will certainly boost our ability to generate prognostic and predictive biomarkers. We have now included text in the Discussion section to highlight these important issues. Specifically, we say *"It should be noted that molecular profiles of tumors used in this study reflect steady-state genomic and transcriptomic states at the time of initial biopsy. Clearly, as patients receive treatments, tumors' microenvironment and molecular compositions may change in response to the interventions. As data on the effects of treatments and their outcomes become available in the future, integrating them will help prioritize biomarkers for clinical utility."*

3 The prognostic value of Module Perturbation Scores varied greatly by Cancer tissue type, being best for Breast and Prostate and of lesser value for other cancers. Why?

We thank the reviewer for raising an important point. We fully agree with the reviewer's comments that the prognostic value of modules vary across cancers. We hypothesize that module perturbations as computed from bulk transcriptomic measurements alone may not be fully reflective of disease trajectories in all cancer contexts. We envision that single-cell measurements that can better reflect the inherent heterogeneity in tumors could boost prognostic signal of module perturbation scores. Furthermore, we believe that capturing genomic,

transcriptomic, epigenetic, proteomic and metabolomic profiles of tumors would provide a more complete view of disease states and clinical trajectories. Specifically, we state in the Discussion section *“Interestingly, we found the contributions of module perturbation scores to survival predictions to be quite variable between cancers (e.g., LGG and HNSC; Fig S28- S30). One potential explanation is that only capturing genomic and/or transcriptomic states of tumors is unlikely to fully capture disease trajectories in all cancer contexts. Developing technologies and building compendia for genomic, transcriptomic, epigenetic, proteomic and metabolomic analyses of patient tumors with matched clinical data is likely to provide a more wholistic view of disease states and clinical outcomes.”*.

4. It is likely that cases sampled were obtained from Tertiary Cancer Centres. This may have biased cases to larger or more advanced cancers. This can be checked by comparing cancer stage with prevalence of each stage in general cancer statistics. The cases may have been more biologically aggressive than average. This could have been assessed by tumor histologic grade that while imperfect, is a measure of aggressiveness.

We thank the reviewer for bringing up a crucial point about potential confounds in the underlying data used in our analyses. To test if the make-up of patients based on cancer stage in TCGA deviates from what is observed in the general population, we compared stage-wise data from TCGA with data from the Surveillance, Epidemiology and End Results (SEER) program. In the SEER cohorts, we defined tumors to be ‘early’ stage when they were labeled ‘localized’ (and ‘in situ’, for bladder cancer) and defined tumors to be ‘advanced’ stage when the tumors were labeled ‘regional’. For the TCGA cohorts, we called tumors at stage I or stage II to be ‘early’ stage and those at stage III or stage IV to be ‘advanced’. Comparisons between TCGA and SEER cohorts were restricted to matched tissues-of-origin cancers in patients presenting with non-metastatic disease at time of diagnosis. We find that in a majority of TCGA cancers examined, patients with advanced disease did not make up the majority (9 of 12 cancers). Secondly, we found that the proportion of TCGA patients with advanced disease were not substantially more than in the general population (11 out of 12 cancers; difference $\leq 25\%$). In fact, the proportions of TCGA patients with early and advanced disease were quite similar to the general population in 8 of the 12 cancers tested (difference $\leq 25\%$). We have described the details of these analyses in the Methods section and included a new subpanel to summarize the results (Figure S12a). We state in the Results section *“In addition to diversity of*

cancer types and the wealth of linked clinical, genomic and molecular data on patients, we used TCGA as our primary discovery cohort since the distribution of pathologic stages are consistent with general population statistics in the United States in multiple cancers (Fig S12a).". The details of these analyses are provided in the Methods section as "We grouped tumors of stage I/IA/IB/II/IIA/IIB/IIC to be 'early' stage and tumors of stage III/IIIA/IIIB/IIIC/IV/IVA/IVB/IVC to be 'advanced' stage (AJCC pathologic stage). In cancers for which population-statistics (<https://seer.cancer.gov/statfacts/>, SEER 1975-2019) were available in tissue-matched cancers, we manually obtained proportion of early (in situ or localized) and advanced stage (regions) cancers in patients presenting with non-metastatic disease at time of diagnosis. We compared the proportions of early and advanced disease in TCGA and SEER cohorts for bladder (BLCA in TCGA), breast (BRCA in TCGA), colorectal (COAD, READ in TCGA), esophageal (ESCA in TCGA), kidney (KIPAN in TCGA), liver (LIHC in TCGA), lung (LUAD, LUSQ in TCGA), melanoma (SKCM in TCGA), oral (HNSC in TCGA), pancreas (PAAD in TCGA), stomach (STAD in TCGA) and thyroid (THCA in TCGA) cancers.". Additionally, to highlight the underlying bias of the patient cohorts used for the discovery of PCMs, we state in the Discussion section "While we believe our overall approach would be highly beneficial for clinical decision-making, the specific predictions generated in this study may be affected by biases that inevitably exist in the underlying data distributions. Their clinical applicability requires further validation on cohorts that closely resemble demographic (e.g., sex, race and ethnicity, age etc.) and tumor characteristics (e.g., histopathology, tumor stage, grade etc.) of the general population. Robust statistical standards, internal cross-validation, and external cross-validation on independent cohorts certainly mitigate some of these concerns, providing confidence that our results may be generalizable.".

Figure S11. Numbers of prognostic cancer modules across different module categories for different cancers. (a) Fraction of patients with early or advanced stage cancer at time of diagnosis from the SEER database and in tissue-matched TCGA cohorts (Methods). Cancers are shown in rows. (b) Number of modules in each module category that specify significant prognostic value for overall survival (black bars) and progression-free interval survival (gray bars). For GO terms and pathways, de novo discovered DNA and RNA modules, numbers refer to non-redundant modules (as defined through Affinity Propagation (Frey and Dueck, 2007)) and the numbers in parentheses, the full set of redundant modules (see Methods). (c) Number of prognostic cancer modules (both overall survival and progression-free interval survival) in each module category (Y-axis; module categories indicated) across different TCGA cohorts (labels on top).

References: Chakravarty, D., Gao, J., Phillips, S., Kundra, R., Zhang, H., Wang, J., Rudolph, J.E., Yaeger, R., Soumerai, T., Nissan, M.H., et al. (2017). OncoKB: A Precision Oncology Knowledge Base. *JCO Precis. Oncol.* 1, 1–16. Cover, T.M., and Thomas, J.A. (2006). *Elements of information theory* (Hoboken, N.J: Wiley-Interscience). Domany, E. (2014). Using High-Throughput Transcriptomic Data for Prognosis: A Critical Overview and Perspectives. *Cancer Res.* 74, 4612–4621. Drier, Y. (2017). pathifier (Bioconductor). Drier, Y., Sheffer, M., and Domany, E. (2013). Pathway-based personalized analysis of cancer. *Proc. Natl. Acad. Sci.* 110, 6388–6393. Frey, B.J., and Dueck, D. (2007). Clustering by Passing Messages Between Data Points. *Science* 315, 972–976. Goodarzi, H., Elemento, O., and Tavazoie, S. (2009). Revealing global regulatory perturbations across human cancers. *Mol. Cell* 36, 900–911. Liberzon, A., Birger, C., Thorvaldsdóttir, H., Ghandi, M., Mesirov, J.P., and Tamayo, P. (2015a). The Molecular Signatures Database Hallmark Gene Set Collection. *Cell Syst.* 1, 417–425. Liberzon, A., Birger, C., Thorvaldsdóttir, H., Ghandi, M., Mesirov, J.P., and Tamayo, P. (2015b). The Molecular Signatures Database Hallmark Gene Set Collection. *Cell Syst.* 1, 417–425. Martin, M.L., Zeng, Z., Adileh, M., Jacobo, A., Li, C., Vakiani, E., Hua, G., Zhang, L., Haimovitz-Friedman, A., Fuks, Z., et al. (2018). Logarithmic expansion of LGR5+ cells in human colorectal cancer. *Cell. Signal.* 42, 97–105. Shaw, K.R.M., and Maitra, A. (2019). The Status and Impact of Clinical Tumor Genome Sequencing. *Annu. Rev. Genomics Hum. Genet.* 20, 413–432. Sheffer, M., Bacolod, M.D., Zuk, O., Giardina, S.F., Pincas, H., Barany, F., Paty, P.B., Gerald, W.L., Notterman, D.A., and Domany, E. (2009). Association of survival and disease progression with chromosomal instability: a genomic exploration of colorectal cancer. *Proc. Natl. Acad. Sci. U. S. A.* 106, 7131–7136.

Referees' report, second round of review

Reviewer 1

Apologies for taking so long to get to this. I was busy, and reading the detailed response of the authors thoroughly was not an easy task.

I thank the authors for taking such care to make their response (relatively) readable, by carefully explaining what was done in response to each point I raised, and where were the resulting changes placed in the new version.

I did not re-read the entire paper, only the parts relevant to my critique of the original version. The authors have addressed all my concerns. As I wrote before, I have a strong bias in favor of characterizing a tumor in terms of biologically

meaningful pathway (or module) based scores, rather than expression vectors of single genes. This paper takes a significant step towards making this approach more credible.

By and large, there seems to be good agreement between the individual scores derived here and by Pathifier. I am neither surprised nor worried by the discrepancies reported here; Pathifier uses the expression signature of normal tissue samples as its reference to characterize a tumor sample, whereas the present method uses the expression levels of all genes as a reference to measure the deviation of the expression of a particular gene and group of genes. Hence no wonder that the results, both on statistical significance of a module and on individual samples, are not in full agreement between the methods. In fact they complement each other.

The fact that one does not need normal samples for this analysis is a considerable advantage of the present method over methods like Pathifier. Perhaps this should be emphasized.

Reviewer 3

The authors have addressed the reviewers' concerns well.

Authors' response to the second round of review

N/a